

# Reconstruction of $\delta^{13}C_{DIC}$ in the Atlantic Ocean: A Probabilistic Machine Learning Approach for Filling Historical Data Gaps

Hui Gao[1,2], Zelun Wu[1], Zhentao Sun[1], Diana Cai[3], Meibing Jin[4], Wei-Jun Cai[1*]

[1]School of Marine Science and Policy, University of Delaware, Newark, Delaware, USA.
5   [2]College of Chemistry and Environmental Science, Guangdong Ocean University, Zhanjiang, China.
[3]Center for Computational Mathematics, Flatiron Institute, New York, New York, USA.
[4]International Arctic Research Center, University of Alaska Fairbanks, Fairbanks, Alaska, USA.

*Correspondence to*: Wei-Jun Cai (wcai@udel.edu)

10   **Abstract.** Stable carbon isotope composition of marine dissolved inorganic carbon (DIC), $\delta^{13}C_{DIC}$, is a valuable tracer for oceanic carbon cycling. However, its observational coverage remains much sparser than that of DIC and other physical or biogeochemical variables, limiting its full potential. Here, we reconstruct $\delta^{13}C_{DIC}$ in the Atlantic Ocean using a probabilistic machine learning framework, Gaussian Process Regression (GPR). We compiled data from 51 historical cruises, including a high-resolution 2023 A16N section, and applied secondary quality control via crossover analysis, retaining 37 cruises for 15   model training, validation, and testing. The trained GPR model achieved an average bias of −0.007 ± 0.082 ‰ and an overall uncertainty of 0.11 ‰, arising from measurement (0.07 ‰), mapping (0.08 ‰), and negligible input-variable ($3.77 \times 10^{-14}$ ‰) errors. Using the GLODAPv2.2023 Atlantic dataset as predictors, the reconstruction expanded the number of acceptable $\delta^{13}C_{DIC}$ samples by a factor of 7.65, from 8,941 to 68,435 across the Atlantic basins. The resulting dataset markedly improves the spatial resolution in longitude, latitude, and depth, and provides enhanced temporal continuity over the past four decades. 20   Compared to the sparse original measurements, the reconstruction reduces spatial discontinuities and reveals finer vertical structures consistent with other high-resolution biogeochemical observations. This reconstructed $\delta^{13}C_{DIC}$ dataset provides new opportunities to resolve regional carbon cycle dynamics, validate Earth system models, refine estimates of oceanic carbon uptake, and extend climate reanalysis records. The data are publicly accessible at the data repository Zenodo under the following DOI: https://doi.org/10.5281/zenodo.16907402 (Gao et al., 2025).



## 1 Introduction

The stable carbon isotope ratio, $\delta^{13}C$, has been widely applied as a tracer in marine carbon research, providing valuable insights into various processes within the oceanic carbon system. Specifically, the $\delta^{13}C$ of dissolved inorganic carbon (DIC), denoted as $\delta^{13}C_{DIC}$ (expressed in per mil, ‰, relative to the VPDB standard), has proven instrumental in studies encompassing estimating rates of biological production in surface ocean mixed layer (Quay et al., 2003, 2009; Yang et al., 2019), quantifying anthropogenic carbon inputs and accumulations in ocean basins (Quay et al., 1992, 2003, 2007, 2017; Körtzinger et al., 2003; Olsen and Ninnemann, 2010; Racapé et al., 2013), and validating earth system models (Sonnerup and Quay, 2012; Schmittner et al., 2013; Liu et al., 2021; Claret et al., 2021), making it an indispensable parameter in understanding the complexities of the marine carbon cycle.

Measurements of $\delta^{13}C_{DIC}$ in the ocean trace their roots to the mid-20th century, with significant advancements occurring in the 1970s and 1980s due to the development of more precise mass spectrometry techniques. A pivotal moment in marine isotope research came with Kroopnick's comprehensive analyses of $\delta^{13}C_{DIC}$ distribution in the Atlantic Ocean (Kroopnick, 1980) and globally (Kroopnick, 1985), which provided critical insights into isotopic patterns across the oceans. Over subsequent decades, research based on the collected observational $\delta^{13}C_{DIC}$ dataset continues to increase (Gruber et al., 1999; Quay et al., 2003; Quay et al. 2017; Schmittner et al., 2013). The creation of databases such as Global Ocean Data Analysis Project (GLODAP) further enhanced access to $\delta^{13}C_{DIC}$ data and other carbon-related parameters (Olsen et al., 2016). However, unlike other carbon datasets such as DIC and total alkalinity (TA), $\delta^{13}C_{DIC}$ lacks secondary quality control until Becker et al. (2016) introduced an internally consistent $\delta^{13}C_{DIC}$ dataset for the North Atlantic, marking a significant step in addressing biases and improving data reliability. But still, unlike other carbon datasets, $\delta^{13}C_{DIC}$ lacks necessary sufficiently high spatial resolution for it to be effective in ocean carbon cycle research and as a tracer for anthropogenic carbon accumulation in the ocean.

Traditionally, $\delta^{13}C_{DIC}$ data have been collected by preserving and transporting seawater samples to shore-based laboratories for analysis using Isotope Ratio Mass Spectrometry (IRMS). Although IRMS approach is highly precise and accurate, it is labor-intensive and unsuitable for at-sea analysis, and unable to simultaneously also measure DIC concentrations. These limitations have significantly restricted the collection of $\delta^{13}C_{DIC}$ data, limiting the ability to capture spatiotemporal variabilities and long-term trends of $\delta^{13}C_{DIC}$. For example, in the Atlantic Ocean, only 6,820 $\delta^{13}C_{DIC}$ measurements were collected across 32 cruises over 40 years, averaging just 213 samples per cruise (Becker et al., 2016). Along the A16N transect, approximately 500 $\delta^{13}C_{DIC}$ samples were collected during the 1993 and 2013 cruises and only 38 surface water samples were collected for $\delta^{13}C_{DIC}$ analysis during the 2003 cruise, compared to over 3,000 DIC samples analysed at sea per cruise. To overcome this severe bottleneck, the Cai Lab has developed a precise, rapid, and sea-going suitable $\delta^{13}C_{DIC}$ analytical method with a precision of better than ±0.05 ‰ based on the Cavity Ring-Down Spectroscopy (CRDS) stable carbon isotope analyzer, G2131-*i* (Su et al., 2019; Deng et al., 2022; Sun et al., 2024; Sun et al., 2025). This method has been extensively tested during several



observations and studies. During a long (58-day) ocean cruise along A16N in 2023, approximately 3,500 $\delta^{13}C_{DIC}$ samples were collected and ~3000 samples were analysed at sea, alongside with DIC observations. This progress has significantly surpassed historical datasets of about 500 samples per cruise in 1993 and 2013 (Sun et al. 2025).

Despite extensive field data collection efforts, observations of $\delta^{13}C_{DIC}$ remain sparse compared to other inorganic carbon chemistry variables (e.g., DIC and TA). The distribution and variations of inorganic carbon chemistry within water masses are governed by local physical and biogeochemical processes, resulting in region-specific stoichiometric relationships among inorganic carbon variables. These relationships are typically nonlinear and exhibit spatial and temporal variability, making it challenging to determine general distribution patterns. Over the past few decades, advancements in machine learning

techniques coupled with accumulated observational data have facilitated numerous studies that interpolate inorganic carbon chemistry variables, in particular the partial pressure of $CO_2$ ($pCO_2$) due to its relatively high spatial coverage, from satellite data and reanalysis products (e.g., Roobert et al., 2024; Wu et al., 2025). These methodological developments present a promising opportunity to investigate the potential for deriving $\delta^{13}C_{DIC}$ data from more abundantly measured variables.

Given the limited and fragmented $\delta^{13}C_{DIC}$ dataset compared to other parameters such as DIC, and to fully utilize the $\delta^{13}C_{DIC}$

tracer approach for quantifying anthropogenic $CO_2$ uptake by the ocean (Quay et al., 2017), the rate of ocean biological production (Esposito et al., 2019; Quay et al., 2009, 2020, 2023), and carbon cycling across the land-ocean interface (Alling et al., 2012; Kwon et al., 2021; Samanta et al., 2015), this study aims to reconstruct a high-resolution $\delta^{13}C_{DIC}$ dataset for the Atlantic Ocean using Gaussian Process Regression (GPR), a probabilistic machine-learning approach capable of capturing nonlinear relationships and spatial-temporal variability. This reconstruction integrates historical $\delta^{13}C_{DIC}$ observations in the

Atlantic Ocean with new high-resolution data collected along the A16N transect in 2023 (Sun et al., 2025), and the product of GLODAPv2.2023 (Lauvset et al., 2024). The final product consists of two components with comprehensive uncertainty analysis: 1) a quality-controlled $\delta^{13}C_{DIC}$ observational dataset compiled from 51 cruises with a crossover analysis using standardized protocols, and 2) a machine learning-reconstructed $\delta^{13}C_{DIC}$ dataset derived from other inorganic carbon chemistry variables. The structure of this study is as follows: **Sect. 2** describes the datasets used, the secondary quality control of $\delta^{13}C_{DIC}$,

and the methodology for reconstructing the $\delta^{13}C_{DIC}$ dataset. **Sect. 3** evaluates the accuracy, performance, and applicability of the reconstructed $\delta^{13}C_{DIC}$ dataset in resolving its spatial and temporal distribution. **Sect. 4** presents the conclusions. **Sect. 5 and 6** provide access the dataset, the codes used for its generation, and the figures presented in this study.



## 2 Data and Methods

### 2.1 Data collection

#### 2.1.1 Historical $\delta^{13}C_{DIC}$ Data Collection

In the Atlantic Ocean, $\delta^{13}C_{DIC}$ data from a total of 51 cruises (**Fig. 1 and Table 1**) were compiled from several international research initiatives, including GLODAP, Ocean Carbon and Acidification Data System (OCADS), CLIVAR and Carbon Hydrographic Data Office (CCHDO) and the internally consistent dataset of $\delta^{13}C_{DIC}$ in the North Atlantic Ocean (NAC13v1, Becker et al., 2016). From the original dataset published by Becker et al. (2016), we excluded four cruises: 35TH20060521,
74JC20120601, 74DI20140606, and OMEX1NA, due to missing essential corresponding parameters, i.e., variables used for model training. The remaining cruises, including those from other sources and the 28 cruises retained from Becker et al. (2016), comprise a total of 51 cruises, covering 369 stations and 15,225 $\delta^{13}C_{DIC}$ samples. To ensure internal consistency, samples from depths greater than 2,000 m were selected for crossover analysis, as deep-water masses at these depths are minimally affected by anthropogenic carbon increases. This criterion resulted in 3,772 samples from 305 stations deeper than 2,000 m (highlighted
as red points in **Fig. 1a**). The temporal and latitudinal distributions of the $\delta^{13}C_{DIC}$ data are illustrated in **Fig. 1b and 1c, respectively**. These datasets span from 1981 to 2023, with the most comprehensive annual $\delta^{13}C_{DIC}$ sampling occurring along the A16N in 2023 (**Fig. 1b**). Geographically, the majority of samples are concentrated in the North Atlantic, particularly between latitudes 25° N and 60° N (**Fig. 1c**). However, the sampling is spatially and temporally uneven, that is, data are sparse in certain years and latitudes, which underscores the need for an approach capable of generating robust predictions and
uncertainty estimates in poorly sampled regions, such as the GPR method applied in this study.



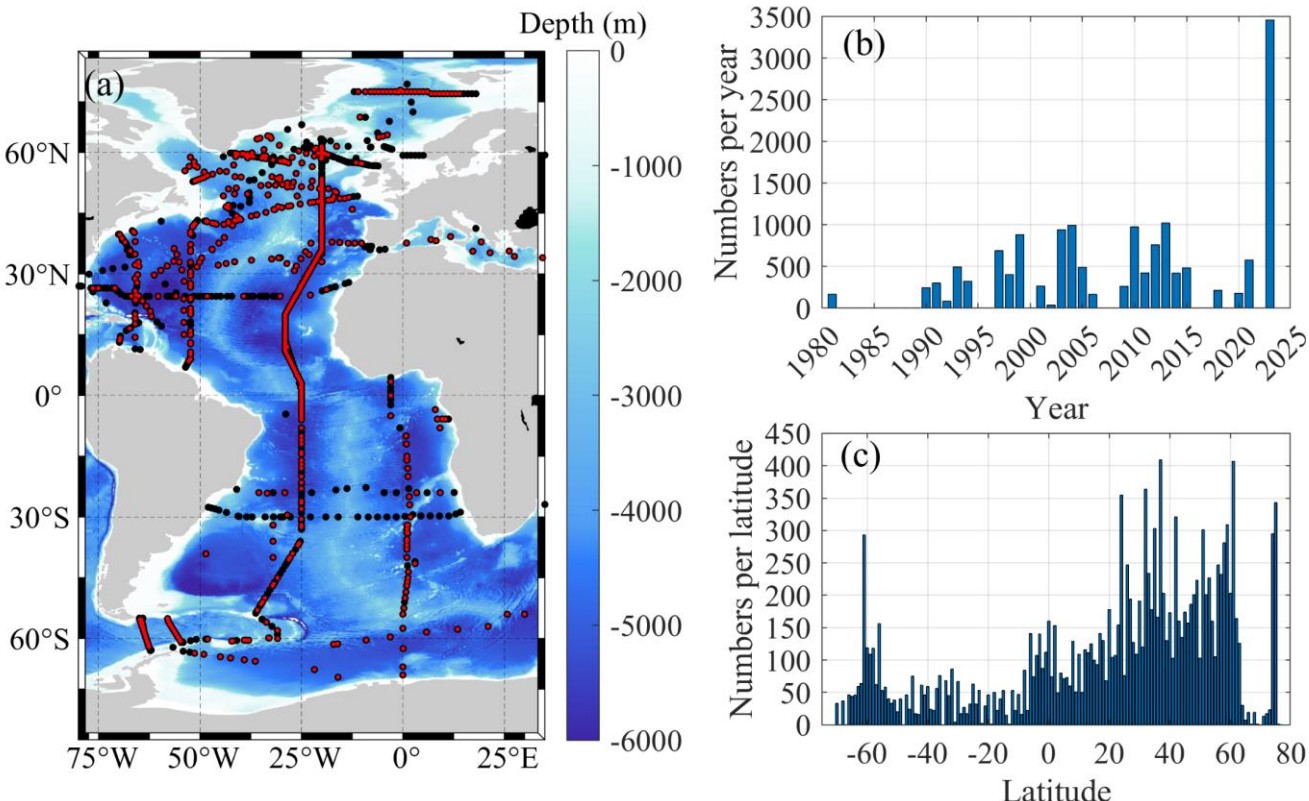

**Figure 1. Overview of the collected $\delta^{13}C_{DIC}$ data.** (a) Map of all stations with $\delta^{13}C_{DIC}$ data in the Atlantic Ocean, with stations containing samples deeper than 2,000 m highlighted in red. (b) Temporal distribution of $\delta^{13}C_{DIC}$ data, organized by year of collection. (c) Total number of $\delta^{13}C_{DIC}$ samples aggregated by each degree of latitude.


**Table 1. Information about Cruises that contains $\delta^{13}C_{DIC}$, and adjustment for each cruise.**

| Cruse No. | Expocode | Cruise Name | Dates | C13 PIs | Adjustments from Becker et al. (2016) | Additional Adjustments |
|---|---|---|---|---|---|---|
| 1 | 06AQ20101128 | A12 SR04 ANT XXVII/2 | 2010-11-28/2011-02-05 | R. Key, A. McNichol, S. Heuven | / | 0 |
| 2 | 06MT19941012 | WOCE A02 06MT30_2 CARINA | 1994-10-12/1994-11-12 | A. Körtzinger, H. Erlenkeuser | −0.07 | 0 |



| 3 | 06MT19970515 | WOCE AR12 AR24 06MT39_2 | 1997-05-15/1997-06-07 | A. Körtzinger, H. Erlenkeuser | 0 | 0 |
|---|---|---|---|---|---|---|
| 4 | 06MT19970611 | A02 06MT39_3 | 1997-06-11/1997-07-03 | A. Körtzinger, H. Erlenkeuser | 0 | 0 |
| 5 | 06MT19970707 | WOCE AR05 AR07 AR07W 06MT39_4 | 1997-07-07/1997-08-07 | A. Körtzinger, H. Erlenkeuser | 0 | 0.05 |
| 6 | 06MT19970815 | WOCE AR07E 06MT39_5 | 1997-08-14/1997-09-09 | A. Körtzinger, H. Erlenkeuser | 0 | 0.05 |
| 7 | 06MT19990610 | 06MT199906 06MT452 06MT45_2 | 1999-06-10/1999-07-09 | A. Körtzinger, H. Erlenkeuser | −0.20 | 0 |
| 8 | 06MT19990711 | AR07 AR07W 06MT45_3 | 1999-07-10/1999-08-11 | A. Körtzinger, H. Erlenkeuser | −0.20 | 0 |
| 9 | 06MT20010507 | A02 CARINA SFB 460; | 2001-05-07/2001-05-31 | A. Körtzinger, H. Erlenkeuser | −0.30 | 0 |
| 10 | 06MT20030723 | CARINA SFB 460; | 2003-07-24/2003-08-26 | A. Körtzinger, H. Erlenkeuser | −0.15 | 0 |
| 11 | 06MT20040311 | CARINA TTO A2 TTO_NAS_2004 | 2004-03-11/2004-04-13 | D. W. R. Wallace, H. Erlenkeuser | 0.10 | 0.03 |
| 12 | 06MT20110405 | M84_3 06ME20110405 MT84_3 Mediterranean Sea | 2011-04-05/2011-04-28 | T. Tanhua, G. Schnaller | / | NC |
| 13 | 18DL20150710 | ArcticNet1502 | 2015-07-10 / 2015-08-20 | A. Mucci | / | NC |
| 14 | 29GD20120910 | (EUROFLEETS) Iberia-Forams | 2012-09-10/2012-09-15 | A.Voelker | / | NC |
| 15 | 316N19810401, 316N19810416, 316N19810516, 316N19810619, 316N19810721, 316N19810821, 316N19810923 | TTO-NAS | 1981-04-01 / 1981-10-19 | C. D. Keeling, P. Guenther | 0 | 0 |
| 16 | 316N19970717 | WOCE A20 | 1997-07-17/1997-08-10 | R. Key | −0.05 | 0 |
| 17 | 316N19970815 | WOCE A22 | 1997-08-15/1997-09-03 | R. Key | NC | 0.05 |





| 18 | 316N20030922 | A20_2003 | 2003-09-22/2003-10-20 | R. Key, P. Quay | NC | NC |
|---|---|---|---|---|---|---|
| 19 | 316N20031023 | A22_2003 | 2003-10-23/2003-11-13 | R. Key, P. Quay | NC | NC |
| 20 | 33AT20120419 | A20_2012 | 2012-04-19/2012-05-15 | R.Key, A.Mcnichol; | / | 0 |
| 21 | 33AT20120324 | A22_2012 | 2012-03-24/2012-04-17 | R.Key, A.McNichol | / | 0 |
| 22 | 33LG20060321 | A21_2006 LMG200603 | 2006-03-21/2006-04-04 | T.Guilderson, P.Quay | / | NC |
| 23 | 33LG20090916 | A21_2009 LMG200909 | 2009-09-16/2009-10-09 | T.Guilderson, P.Quay | / | NC |
| 24 | 33MW19910711 | A16S SATL-91 OACES91 | 1991-07-11/1991-08-05 | P. Quay, R. Key | 0 | 0.04 |
| 25 | 33MW19930704 | A16N AR21 OACES93 NATL-93 | 1993-07-04/1993-08-30 | P. Quay, R. Key | C | C |
| 26 | 33RO19980123 | A05 AR01 | 1998-01-23/1998-02-24 | R. Key, P. Quay | NC | 0.03 |
| 27 | 33RO20030604 | A16N_2003 | 2003-06-04/2003-08-11 | A. McNichol | / | NC |
| 28 | 33RO20050111 | A16S A23 | 2005-01-11/2005-02-24 | A. McNichol | / | −0.10 |
| 29 | 33RO20100308 | A13.5 | 2010-03-08/2010-04-18 | R. Key, A. McNichol | / | 0 |
| 30 | 33RO20110926 | A10 | 2011-09-26/2011-10-31 | A. Foreman, A. Coppola | / | NC |
| 31 | 33RO20130803 | A16N_2013 | 2013-08-03/2013-10-03 | A. McNichol | / | 0 |
| 32 | 33RO20131223 | A16S A23 | 2013-12-23/2014-02-04 | A. McNichol, R. Key | / | 0 |
| 33 | 33RO20200321 | A12 A13.5 | 2020-03-21/2020-04-17 | W-J. Cai | / | 0.07 |
| 34 | 33RO20230306 | A16N_2023, Leg 1 | 2023-03-06/2023-04-07 | W-J. Cai | / | 0 |



| 35 | 33RO20230413 | A16N_2023, Leg 2 | 2023-04-13/2023-05-09 | W-J. Cai | / | 0 |
|---|---|---|---|---|---|---|
| 36 | 35A320031214 | BIOZAIRE_III | 2003-12-14/2004-01-07 | A. Vangriesheim | / | NC |
| 37 | 35TH20020611 | Ovide02 A25 | 2002-06-10/2002-07-12 | H. Mercier | 0.25 | 0 |
| 38 | 49NZ20031106 | A10 | 2003-11-06/2003-12-05 | Y. Kumamoto | / | NC |
| 39 | 58GS20030922 | GO-SHIP A75N | 2003-09-22 / 2003-10-13 | A. Olsen | NC | 0 |
| 40 | 58GS20130717 | CLIVAR_75N_2013 | 2013-07-17 / 2013-07-30 | A. Olsen | / | 0 |
| 41 | 58GS20150410 | A01 AR07E_2015 | 2015-04-10/2015-04-26 | A. Olsen & U. Ninnemann | / | 0.18 |
| 42 | 58JH19920712 | CARINA WOCE AR18b | 1992-07-12 / 1992-07-28 | R. Nydal | NC | 0 |
| 43 | 58JH19940723 | CARINA WOCE AR18d | 1994-07-23 / 1994-08-16 | R. Nydal | NC | 0 |
| 44 | 64TR19900417 | CARINA A16N | 1990-04-17 / 1990-05-31 | S. Wijma | poor | −0.76 |
| 45 | 740H20081226 | A12 A21 | 2008-12-26/2009-01-30 | W. Jenkins | / | 0 |
| 46 | 740H20180228 | A9.5 A10 A09 | 2018-02-28/2018-04-10 | H. Graven | / | NC |
| 47 | 74DI20120731 | EEL_2012_D379 JR271_D379_2012 | 2012-07-31 /2012-08-17 | A. M. Griffiths, M. P. Humphreys, E. P. Achterberg | 0 | 0 |
| 48 | 74JC20100319 | A21 A23 | 2010-03-19/2010-04-24 | W.Jenkins | / | 0 |
| 49 | 74JC20181103 | SR01B | 2018-11-03/2018-11-22 | R.Key, A. McNichol | / | NC |
| 50 | 325020210316 | A20_2021 | 2021-03-16/2021-04-16 | R. Sonnerup, R. Hansman | / | 0 |
| 51 | 325020210420 | A22_2021 | 2021-04-20/2021-05-16 | R. Sonnerup, R. Hansman | / | 0 |



Note: NC denotes cruises that were not considered for the adjustment due to the absence of statistically significant crossovers. The core cruise, identified as the reference, is marked with a "C."

### 2.1.2 $\delta^{13}C_{DIC}$ Data Collection Along A16N in 2023

The A16N cruise in 2023 achieved extensive collection and high-resolution analysis of $\delta^{13}C_{DIC}$ and DIC datasets over two legs using onboard analytical techniques. Samples were collected from CTD rosette bottles into 250 mL borosilicate glass bottles following PICES best practices, preserved with $HgCl_2$, and sealed to prevent biological activity. Analytical measurements were conducted onboard using two coupled systems comprising a $CO_2$ extraction device (AS-D1 $\delta^{13}C_{DIC}$ Analyzer) and a CRDS (Picarro G2131-i), which simultaneously measured DIC concentrations and $\delta^{13}C_{DIC}$ values with high precision. Quality

control measures included frequent calibration using CRMs and homemade standards verified by IRMS, ensuring high data reliability with deviations mostly within ±0.03 ‰ for $\delta^{13}C_{DIC}$ (Sun et al., 2025). This cruise collected approximately 3,500 $\delta^{13}C_{DIC}$ samples along the A16N transect, far exceeding historical datasets of ~500 samples per cruise, with synchronized DIC and $\delta^{13}C_{DIC}$ sampling providing robust datasets for analyzing carbon system dynamics.

### 2.2 Standardized Protocols for Crossover analysis

Secondary quality control (QC) of $\delta^{13}C_{DIC}$ data through crossover analysis ensures consistency across multi-cruise datasets (Becker et al., 2016; Tanhua et al., 2010; Lauvset & Tanhua, 2015). The analysis involves compiling data from overlapping regions, identifying crossover points within 222 km, and comparing $\delta^{13}C_{DIC}$ profiles in deep waters (> 2,000 m) where variability is minimal. Profiles are interpolated to standard depths, and mean offsets are calculated. Systematic biases are identified using least squares minimization, and adjustments are proposed to align datasets without erasing real temporal or

spatial trends. Adjustments are validated against known regional patterns and applied only if they meet accuracy criteria. All steps and corrections are documented to ensure transparency, resulting in reliable $\delta^{13}C_{DIC}$ datasets for carbon cycle analysis. Building on the NAC13v1 dataset provided by Becker et al. (2016), we propose additional adjustment recommendations for these cruises (**Table 1**). Given that cruise 64TR19900417 crossovers with cruise 33MW19930704, 06MT20040311, 33RO20130803, 33RO20230413, showing a very high mean offset and standard deviation −0.76 ± 0.23 ‰, and its $\delta^{13}C_{DIC}$

data is marked as NaN (missing values) in the NAC13v1 dataset, it is excluded from our analysis. After applying additional adjustments, the $\delta^{13}C_{DIC}$ data for the remaining 37 cruises, excluding the 13 cruises without crossover stations (Table 1) and cruise 64TR19900417, exhibit high internal consistency. The internal accuracy of the adjusted $\delta^{13}C_{DIC}$ is determined to be $-4.15\times10^{-5}$ ‰ based on the calculation from Tanhua et al. (2010) and Becker et al. (2016). Finally, a total of 11,950 samples are used to train and test the model (**Fig. 2**). These samples were selected based on the quality flags of the relevant variables,

where only data marked with quality flag values of 2 (acceptable measurement) or 6 (median of replicate measurements) were included, ensuring that the dataset is reliable and suitable for model development.



### 2.3 Model design

Predicting $\delta^{13}C_{DIC}$ in the ocean requires a method that can handle complex, nonlinear relationships and provide reliable uncertainty estimates for scientific interpretation. GPR (Seeger, 2004; Rasmussen & Williams, 2006) is particularly well suited
to this task. As a non-parametric, probabilistic model, GPR not only produces point predictions but also quantifies uncertainty through credible intervals around the estimates. The versatility of GPR arises from its kernel function, which allows prior knowledge about the expected smoothness or variability of the $\delta^{13}C_{DIC}$-environment relationship to be incorporated into the model, making GPR a powerful tool for robust predictions in oceanographic applications.

We employed GPR with a Matern 5/2 kernel as the primary method for all subsequent $\delta^{13}C_{DIC}$ reconstructions. The Matern
class of kernels control function smoothness through its smoothness parameter. Specifically, the 5/2 kernel yields functions that are smooth yet not overly restrictive, offering a balanced representation that aligns well with the expected variability of many data sets. Compared with the widely used squared exponential kernel, which assumes infinitely differentiable, and often unrealistically smooth functions, the Matern 5/2 kernel allows for more plausible modeling of natural variability.

To evaluate this approach's performance, we compared the Matern 5/2 GPR with a suite of alternative regression models,
including GPR with other kernels, as well as additional baselines such as neural networks, support vector regression, and decision trees. The dataset was randomly split into a training set (80%) and a validation set (20%), with model training and hyperparameter tuning performed using 10-fold cross-validation within the training set to mitigate overfitting. An independent test set was reserved for final performance evaluation. Predictive performance was assessed using the Root Mean Squared Error (RMSE) and the coefficient of determination ($R^2$), computed separately for the validation and test sets. Among all tested
models, including GPR with the squared exponential and other kernels (Table 2), GPR with the Matern 5/2 kernel achieved the best predictive performance (lowest RMSE and highest $R^2$) on the validation set as well as the independent test set, while also providing meaningful uncertainty estimates.

**Table 2. Selection of machine-learning models based on Root Mean Squared Error (RMSE), and the coefficient of determination ($R^2$).**

| Model Name | RMSE (Validation) | $R^2$ (Validation) | RMSE (Test) | $R^2$ (Test) |
|---|---|---|---|---|
| Matern 5/2 Gaussian Process Regression | 0.084 | 0.92 | 0.078 | 0.95 |
| Rational Quadratic Gaussian Process Regression | 0.084 | 0.92 | 0.081 | 0.95 |
| Exponential Gaussian Process Regression | 0.084 | 0.92 | 0.079 | 0.95 |
| Squared Exponential Gaussian Process Regression | 0.085 | 0.91 | 0.082 | 0.95 |
| Wide Neural Network | 0.089 | 0.90 | 0.136 | 0.86 |
| Ensemble Bagged Trees | 0.090 | 0.90 | 0.114 | 0.90 |
| Medium Gaussian SVM | 0.093 | 0.90 | 0.071 | 0.96 |
| SVM Kernel | 0.095 | 0.89 | 0.141 | 0.85 |
| Medium Neural Network | 0.094 | 0.89 | 0.091 | 0.94 |



| | | | | |
|---|---|---|---|---|
| Least Squares Regression Kernel | 0.099 | 0.88 | 0.106 | 0.92 |
| Bilayered Neural Network | 0.104 | 0.87 | 0.093 | 0.94 |
| Fine Gaussian SVM | 0.105 | 0.87 | 0.258 | 0.50 |
| Narrow Neural Network | 0.106 | 0.86 | 0.097 | 0.93 |
| Medium Tree | 0.106 | 0.86 | 0.142 | 0.85 |
| Fine Tree | 0.108 | 0.86 | 0.144 | 0.84 |
| Trilayered Neural Network | 0.097 | 0.88 | 0.086 | 0.94 |
| Coarse Tree | 0.117 | 0.83 | 0.122 | 0.89 |
| Ensemble Boosted Trees | 0.119 | 0.83 | 0.163 | 0.80 |
| Coarse Gaussian SVM | 0.123 | 0.82 | 0.113 | 0.90 |
| Stepwise Linear Regression | 0.109 | 0.86 | 0.095 | 0.93 |
| Interactions Linear Regression | 0.109 | 0.85 | 0.099 | 0.93 |
| Quadratic SVM | 0.109 | 0.86 | 0.082 | 0.95 |
| Efficient Linear Least Squares | 0.154 | 0.71 | 0.179 | 0.76 |
| Efficient Linear SVM | 0.157 | 0.70 | 0.203 | 0.69 |
| Linear SVM | 0.156 | 0.70 | 0.201 | 0.70 |
| Robust Linear Regression | 0.157 | 0.70 | 0.208 | 0.68 |
| Cubic SVM | 0.094 | 0.89 | 0.078 | 0.95 |
| Neural Network with the Levenberg-Marquardt algorithm | 0.100 | 0.88 | 0.082 | 0.94 |
| Neural Network with the Bayesian regularization algorithm | 0.089 | 0.90 | 0.084 | 0.94 |
| Neural Network with the scaled conjugate gradient algorithm | 0.122 | 0.81 | 0.118 | 0.90 |

The process of developing and reconstructing the $\delta^{13}C_{DIC}$ data product is outlined in **Fig. 2**. Data collection and preprocessing were conducted following the procedures detailed in **Sect. 2.1** and **Sect. 2.2**. To ensure data reliability, we excluded 13 cruises lacking deep-water crossover stations, and a biased cruise (64TR19900417), as their uncertainties could not be objectively quantified. Collectively, these excluded cruises accounted for less than 3 % of total $\delta^{13}C_{DIC}$ measurements.

The remaining 37 cruises were used for model development, validation and test. Two of them (33RO20050111 and
33MW19930704), from the South and North Atlantic respectively, were randomly selected to form an independent test set (X1). The other 35 cruises formed dataset X2, which was further randomly split into a training set (80 %) and a validation set (20 %). The validation set was used to fine-tune hyperparameters and assess model performance, ensuring generalizability and helping identify overfitting (Wu et al., 2025). The model was trained using paired input variables: longitude, latitude, depth, temperature (T), salinity (S), apparent oxygen utilization (AOU), nitrate (N), silicate (Si), DIC, and atmospheric $CO_2$ ($xCO_2$),
along with corresponding $\delta^{13}C_{DIC}$ values as the target variable (eq. (1)).

$$\delta^{13}C_{DIC} = f(lon, lat, depth, T, S, AOU, N, Si, DIC, xCO_2) \qquad (1)$$

These input variables were selected to comprehensively capture the key physical, biological, and geochemical drivers influencing $\delta^{13}C_{DIC}$. Specifically, longitude, latitude, and depth represent the spatial location of each observation, which is essential for resolving regional and vertical patterns. T and S reflect thermohaline forcing, i.e., the physical processes such as
mixing, stratification, and water mass formation that impact carbon cycling. AOU, nitrate, and silicate are indicators of





biological forcing, as they are influenced by biological productivity, remineralization, and nutrient utilization. DIC is directly related to $\delta^{13}C_{DIC}$, since $\delta^{13}C_{DIC}$ reflects the stable carbon isotopic composition of the DIC pool; thus, variations in DIC are closely tied to changes in $\delta^{13}C_{DIC}$. However, AOU, nutrients, and DIC also reflect partially physics as they are being mixed by ocean circulation. Finally, $xCO_2$ represents external perturbations from air-sea $CO_2$ exchange, which can alter both the
concentration and isotopic composition of DIC.

The independent test set (X1), excluded entirely from training and validation, provides a final evaluation of model performance. Its role in assessing predictive skill across spatial data gaps ensures robust generalization. Finally, the trained GPR model is applied to the full set of hydrographic parameters from the GLODAPv2.2023 Atlantic dataset (https://glodap.info/index.php/merged-and-adjusted-data-product-v2-2023/) to reconstruct a basin-wide $\delta^{13}C_{DIC}$ distribution
across the Atlantic Ocean.

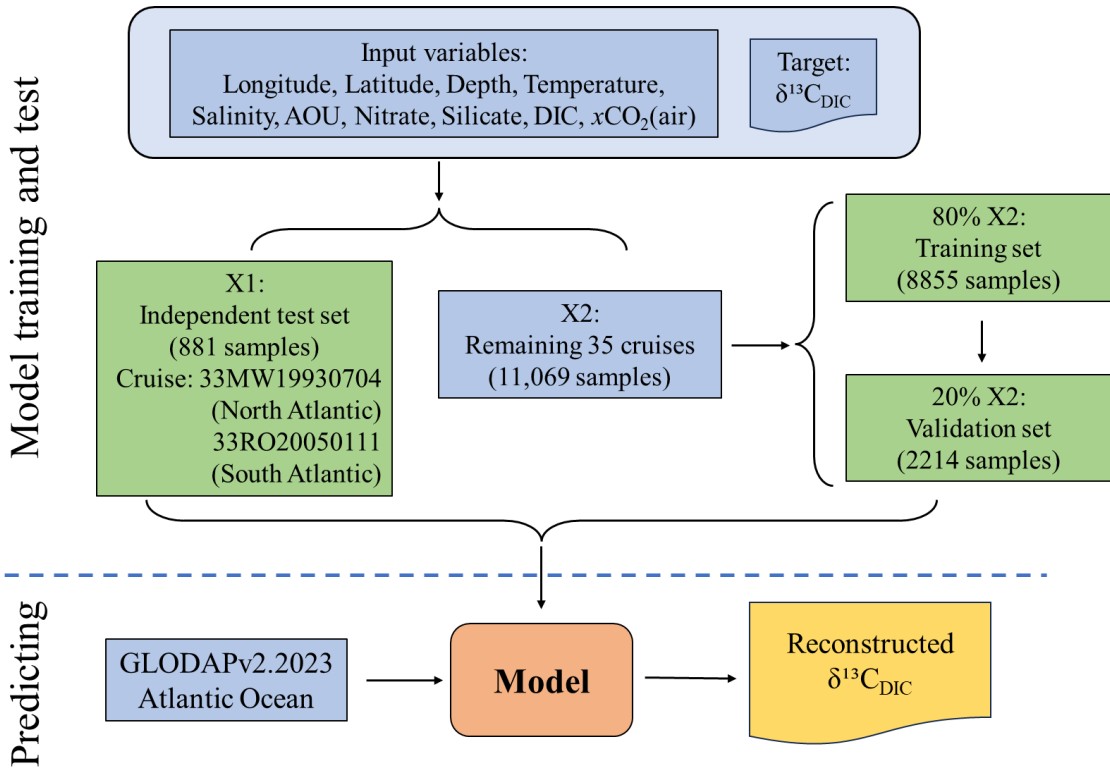

**Figure 2. A flowchart illustrating the machine-learning regression model for reconstructing the $\delta^{13}C_{DIC}$ product.** The blue boxes represent the input, and the yellow box represents output datasets, while the green boxes depict the model training, validation, and independent testing processes. The orange box indicates the final trained model used for prediction.





**2.4 Evaluation of model**

The accuracy of the model outputs was evaluated using various statistical metrics, including the $R^2$, RMSE, mean absolute error (MAE), and mean bias error (MBE). These metrics were calculated for the training, validation, and independent test phases, as defined below:

$$R^2 = 1 - \frac{\sum_{i=1}^{N}(y_{obs,i} - y_{est,i})^2}{\sum_{i=1}^{N}(y_{obs,i} - \overline{y_{obs}})^2} \qquad (2)$$

$$RMSE = \sqrt{\frac{1}{N}\sum_{i=1}^{N}(y_{obs,i} - y_{est,i})^2} \qquad (3)$$

$$MAE = \frac{1}{N}\sum_{i=1}^{N}|y_{obs,i} - y_{est,i}| \qquad (4)$$

$$MBE = \frac{1}{N}\sum_{i=1}^{N}(y_{obs,i} - y_{est,i}) \qquad (5)$$

Here, $i$ represents the $i$-th sample, $y_{obs,i}$ refers to the observed $\delta^{13}C_{DIC}$, $\overline{y_{obs}}$ is the mean of the observed $\delta^{13}C_{DIC}$ values, $y_{est,i}$ denotes the predicted $\delta^{13}C_{DIC}$ values from the final model, and N is the total number of matched samples.

**2.5 Uncertainty of the reconstructed $\delta^{13}C_{DIC}$**

The uncertainty of the reconstructed $\delta^{13}C_{DIC}$ was accumulated from three sources of uncertainties: the direct $\delta^{13}C_{DIC}$ measurement uncertainty from observations ($u_{obs}$), the uncertainty accumulated from the input variables ($u_{inputs}$), and the uncertainty induced by the mapping function ($u_{map}$).

The observational uncertainty $u_{obs}$ inherent in $\delta^{13}C_{DIC}$ measurements varies by analytical method. For samples, when it is 205    analyzed using IRMS, reported uncertainties range from ±0.12 ‰ (Gruber et al., 1999) to ±0.03 ‰ (Quay et al., 2003). For CRDS analysis, uncertainties are reported as ±0.07 ‰ for cruise 33RO20200321 (Gao et al., 2024) and ±0.03 ‰ for cruises 33RO20230306 and 33RO20230413 (Sun et al., 2025). Taking a conservative approach, we adopted an average $u_{obs}$ value of 0.07 ‰.

The input variable uncertainty ($u_{inputs}$) accounts for uncertainties in temperature, salinity, nitrate, silicate, DIC, alkalinity, and 210    $x$CO$_2$. Monte Carlo simulation is performed 1000 times to quantify $u_{inputs}$. Following Carter et al. (2024) and Wu et al. (2025), the perturbation of 0.002, 0.002, 2, 0.4, 0.4, 2 and 0.2 are randomly perturbs to temperature, salinity, AOU, nitrate, silicate, DIC, and $x$CO$_2$. We then recalculated $\delta^{13}C_{DIC}$ using these perturbed inputs and quantified the resulting changes. The uncertainty contribution from each variable was determined as the standard deviation of the differences between the original reconstructed



$\delta^{13}C_{DIC}$ and the noise-perturbed values. The final $u_{inputs}$ was calculated as the square root of the quadratic sum of these individual

uncertainties.

$$u_{inputs}^2 = u_T^2 + u_S^2 + u_{AOU}^2 + u_N^2 + u_{Si}^2 + u_{DIC}^2 + u_{xCO_2}^2 \qquad (6)$$

The uncertainty associated with the mapping function, $u_{map}$, is introduced within the GPR model. GPR uncertainty is quantified as the posterior predictive standard deviation (i.e., the square root of the posterior predictive variance), with a mean value of 0.08 ‰ across the prediction set. Wu et al. (2025) suggested that $u_{map}$ may alternatively be estimated as the RMSE between

reconstructed and observed $\delta^{13}C_{DIC}$ on the training dataset. In our analysis this RMSE closely matches the GPR-derived predictive standard deviation. For consistency and comparability with previous RMSE-based studies, we therefore adopt the RMSE-based definition of $u_{map}$ in this work, while noting that the GPR posterior predictive standard deviation constitutes a conceptually direct and quantitatively similar alternative measure of reconstruction uncertainty.

Assuming independence between these uncertainty sources, the total uncertainty of our estimated $\delta^{13}C_{DIC}$ product, $u_{\delta^{13}C_{DIC}}$,

was determined using the error propagation (Hughes and Hase, 2010; Taylor, 1997):

$$u_{\delta^{13}C_{DIC}} = \sqrt{u_{obs}^2 + u_{inputs}^2 + u_{map}^2} \qquad (7)$$

## 3 Results and Discussion

### 3.1 Evaluation of the GPR model performance

The trained GPR model exhibited robust performance and high accuracy in representing $\delta^{13}C_{DIC}$ characteristics across the

Atlantic Ocean (**Fig. 3**). During the training phase, leveraging a 10-fold cross-validation approach, the model achieved an $R^2$ of 0.92, an RMSE of 0.083 ‰, an MAE of 0.056 ‰, and an MBE of −0.0003 ‰ (**Fig. 3a**). Notably, performance metrics during the validation phase (**Fig. 3b**) were comparable to those in the training phase, demonstrating the model's strong generalization ability, i.e., its capacity to accurately predict data not used for training, and confirming its resistance to overfitting. To further assess the model's generalizability and robustness, an independent test was conducted using $\delta^{13}C_{DIC}$

data from cruises 33MW19930704 and 33RO20050111, ensuring that all samples were entirely independent from the training and validation datasets. In this testing phase, the model maintained high accuracy ($R^2$ = 0.95, RMSE = 0.082 ‰, MAE = 0.056 ‰, MBE = 0.007 ‰, **Fig. 3c**), with most samples clustered closely around the 1:1 line. These results indicate that the model can reliably predict $\delta^{13}C_{DIC}$ across diverse, unobserved spatial and temporal scales. Collectively, these findings demonstrate that the GPR-based $\delta^{13}C_{DIC}$ reconstruction method is highly generalizable and robust, enabling it to accurately capture the

characteristics in $\delta^{13}C_{DIC}$ and provide reliable predictions across the Atlantic Ocean.





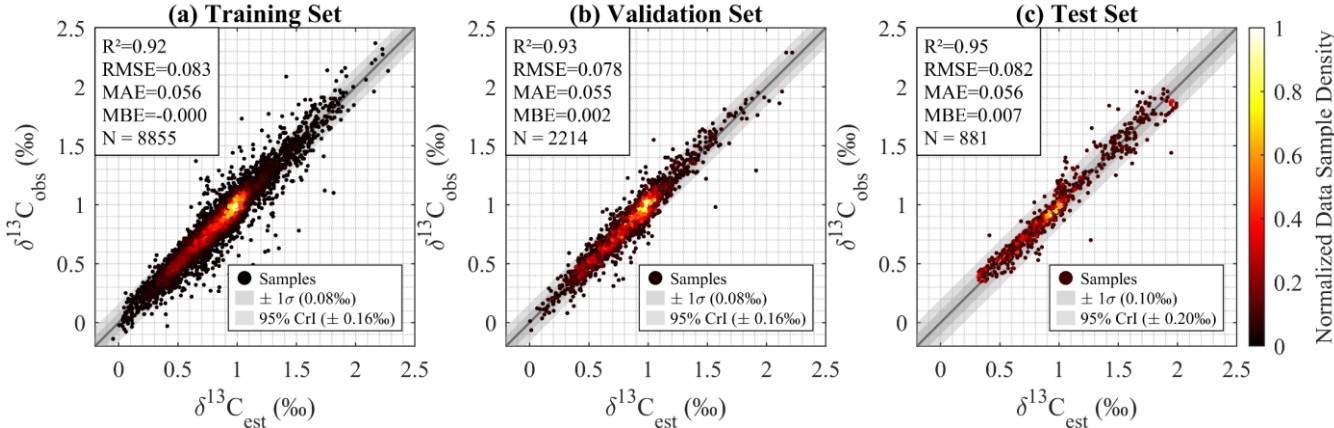

**Figure 3. Regression model evaluation for δ¹³C$_{DIC}$ reconstruction:** Density scatter plots comparing model-estimated (δ¹³C$_{est}$) versus in-situ observed (δ¹³C$_{obs}$) δ¹³C$_{DIC}$ values during (a) training (80 % samples from 35 cruises, with 10-fold cross-validation), (b) validation (20 % samples from 35 cruises), and (c) independent testing (samples from cruises 33RO20050111 and 33MW19930704). Statistical metrics include coefficient of determination (R²), root-mean-square error (RMSE), mean absolute error (MAE), mean bias error (MBE), and sample size (N). Color indicates normalized local data point density within 2D bins. Shaded bands represent predictive uncertainty, with the darker gray band showing one standard deviation (±1σ) and the lighter gray band showing the 95 % credible interval (CrI).

## 3.2 Evaluation of the spatial distribution of δ¹³C$_{DIC}$

The distribution of δ¹³C$_{DIC}$ from independent test cruises 33MW19930704 and 33RO20050111 is used to evaluate the product's capability to capture the spatial patterns of δ¹³C$_{DIC}$ and to quantify the bias (**Fig. 4**). Both cruises are part of the repeated observation section A16, specifically A16N (1993) and A16S (2005), which traverses the entire Atlantic Ocean from the sub-Arctic to the Southern Ocean (**Fig. 4a**). This section has been frequently used in existing research and textbooks to represent Atlantic-scale distributions of carbonate chemistry and other characteristics (i.e., Wanninkhof et al., 2010; Eide et al., 2017; Millero 2013), thus it is selected for evaluating the product's regional applicability.

The spatial patterns of model-estimated (δ¹³C$_{est}$) along cruises 33MW19930704 and 33RO20050111 effectively captured the distributional characteristics of the in-situ observed (δ¹³C$_{obs}$) (**Fig. 4b vs. 4c**). The reconstructed δ¹³C$_{DIC}$ product exhibited a very low section-mean bias of −0.007 ‰ with a standard deviation of 0.082 ‰ (**Fig. 4d**), highlighting the product's reliability in estimating the spatial distribution of δ¹³C$_{DIC}$. Spatially, the discrepancies are very small in most regions of the section, but relatively lager in subpolar regions (around 50° S or 50° N) and vertically in the upper 500 m and below 3000 m (**Fig. 4d**).

Earth System
Science
Data

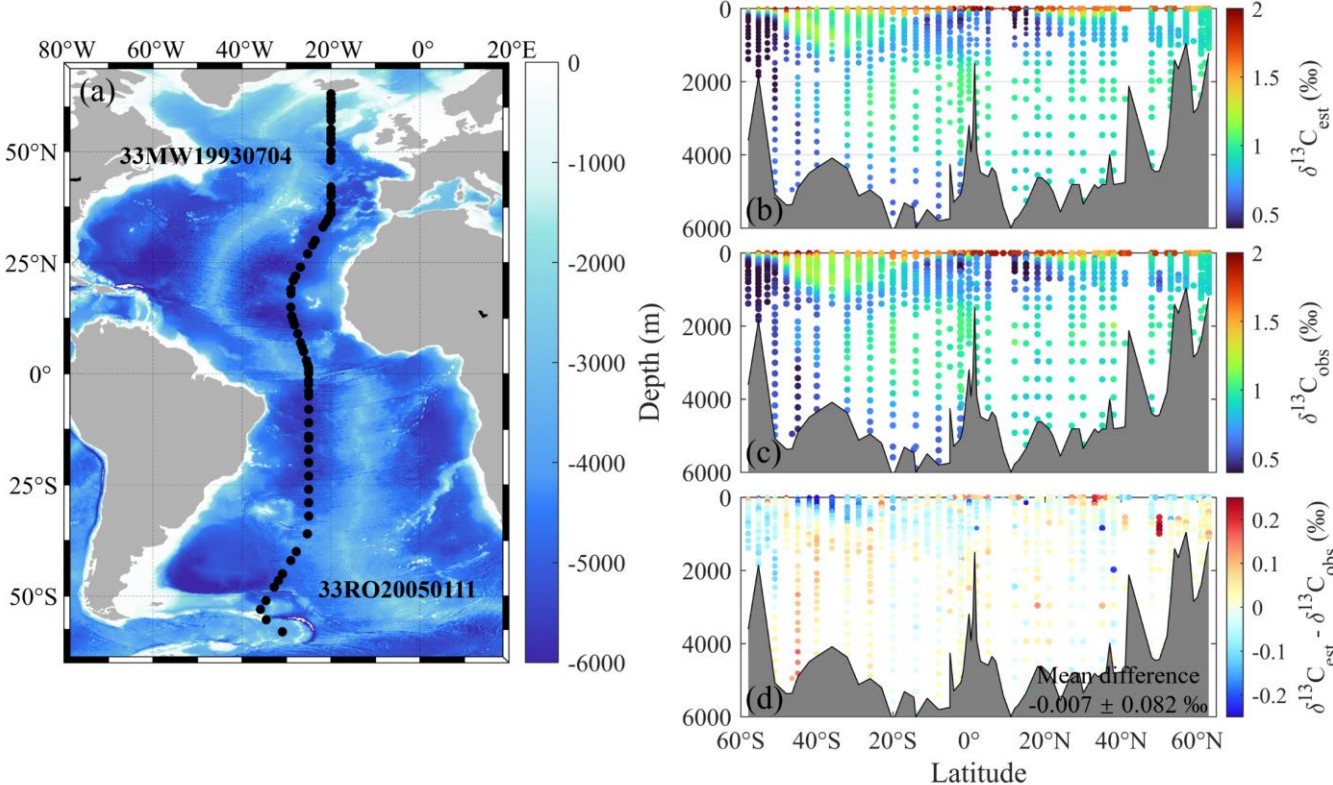

**Figure 4. (a) Station locations of independent test cruises 33MW19930704 and 33RO20050111; Depth profile of (b) model-estimated ($\delta^{13}C_{est}$) and (c) in-situ observed ($\delta^{13}C_{obs}$) $\delta^{13}C_{DIC}$ along cruises 33MW19930704 and 33RO20050111, and (d) spatial distribution of mean bias error (MBE) between $\delta^{13}C_{est}$ and $\delta^{13}C_{obs}$ for the two cruises.** Positive MBE values (red) denote product overestimation, while negative values (blue) indicate underestimation relative to observation. The overall mean difference is $-0.007 \pm 0.082$ ‰.

### 3.3 Evaluation of the product's uncertainty

The uncertainty of the reconstructed $\delta^{13}C_{DIC}$ product was estimated by propagating uncertainties from three primary sources: measurement ($u_{obs}$), input variables ($u_{inputs}$) and mapping ($u_{map}$). Detailed calculations are described in **Sect. 2.5** of the Methods. To ensure a conservative estimate, we adopted a uniform $u_{obs}$ of 0.07 ‰ for all data points. $u_{map}$ was derived from the RMSE between $\delta^{13}C_{est}$ and $\delta^{13}C_{obs}$ in the training set following previous literature (Wu et al., 2025; Roobaert et al., 2024; Sharp et al., 2022), yielding a value of 0.08 ‰. $u_{inputs}$ was quantified via Monte Carlo simulation, considering contributions from seven environmental variables: T, S, AOU, N, Si, DIC, xCO₂. Notably, uncertainties from input variables had a negligible impact, with $u_{inputs}$ estimated at $3.77 \times 10^{-14}$ ‰. Overall, the reconstructed $\delta^{13}C_{DIC}$ product exhibits an average uncertainty of 0.11 ‰ for the entire North Atlantic Ocean. This uncertainty is considered reasonable given the conservative estimation approach, highlighting the product's reliability for $\delta^{13}C_{DIC}$ characterization.



### 3.4 Reconstruction of $\delta^{13}C_{DIC}$

Here, the trained GPR model is applied to GLODAPv2.2023 hydrographic data for $\delta^{13}C_{DIC}$ reconstruction in the Atlantic Ocean. The GLODAPv2.2023 Atlantic dataset comprises 500,137 samples, of which 8,941 contain acceptable $\delta^{13}C_{DIC}$ observations (quality flags = 2 or 6). A total of 124,643 $\delta^{13}C_{DIC}$ values can be reconstructed based on the availability of all required predictor variables (salinity, AOU, Nitrate, Silicate, and DIC). Among these, 68,435 reconstructions are considered high quality, as all six input variables have acceptable quality flags, which is approximately 7.65 times larger than the number of $\delta^{13}C_{DIC}$ observations. The remaining 56,208 reconstructions are based on input variables with unknown or lower quality and are thus assigned a quality flag of 3 (questionable). All 124,643 reconstructions are provided in the Supplementary Dataset, of which the questionable $\delta^{13}C_{DIC}$ samples are provided for transparency but are not recommended for routine use without further quality assessment. The following analyses and figures are restricted to the 68,435 acceptable samples to ensure the robustness of the results unless otherwise noted.

The observed and reconstructed datasets share an intersection of 5,997 samples that have both acceptable $\delta^{13}C_{DIC}$ observations and acceptable input variables for reconstruction. These overlapping samples are used for direct model evaluation (**Fig. 5a**). The comparison between observed and reconstructed $\delta^{13}C_{DIC}$ values yields a high correlation coefficient ($R^2 = 0.89$), with RMSE = 0.094 ‰, MAE = 0.067 ‰, and MBE = −0.009 ‰. The mean difference between reconstruction and observation is −0.009 ± 0.097 ‰. To compare the statistical characteristics of observed and reconstructed $\delta^{13}C_{DIC}$ values, we calculated Gaussian kernel density estimations (KDEs) (**Fig. 5b**). KDEs is a non-parametric method for approximating probability density functions by summing kernel functions (here Gaussian) centered at each data point (Silverman, 1986), producing a continuous estimate that is less sensitive to arbitrary bin edges. For this analysis, KDEs were evaluated on a uniform grid spanning the full $\delta^{13}C_{DIC}$ range, with a bandwidth of 0.1 ‰ chosen to balance smoothing and resolution. No additional jittering was applied as the data contain sufficient variability to avoid overlapping artifacts. The observed (N = 8,941, range from −0.37 to 2.37 ‰) and reconstructed (N = 68,435, range from −0.11 to 2.36 ‰) $\delta^{13}C_{DIC}$ show similar Gaussian KDE curves, exhibiting unimodal distributions, with a primary peak around 1 ‰. Notably, the reconstructed distribution exhibits higher probability density than the observations for $\delta^{13}C_{DIC}$ values between approximately 0.8 ‰ and 1.1 ‰. This difference arises not only from the larger sample numbers but also from the reconstructed distribution being more concentrated, as reflected by a sharper peak and reduced variance. These characteristics likely result from model smoothing and from the much larger, more spatially continuous reconstructed dataset, which together reduce sampling noise. Consequently, the reconstructed values display a slightly sharper central peak and narrower tails than the observations, indicating a tendency of the model to smooth extreme values. The KDE curves decline rapidly to near zero at the extremes, below approximately 0 ‰ and above approximately 2 ‰, indicating very few $\delta^{13}C_{DIC}$ values in those ranges. Overall, the KDE analysis, based on a systematic bandwidth selection and



305    grid evaluation, confirms that the GPR model captures the central tendency and overall shape of the observed $\delta^{13}C_{DIC}$

distribution while producing a somewhat more concentrated estimate.

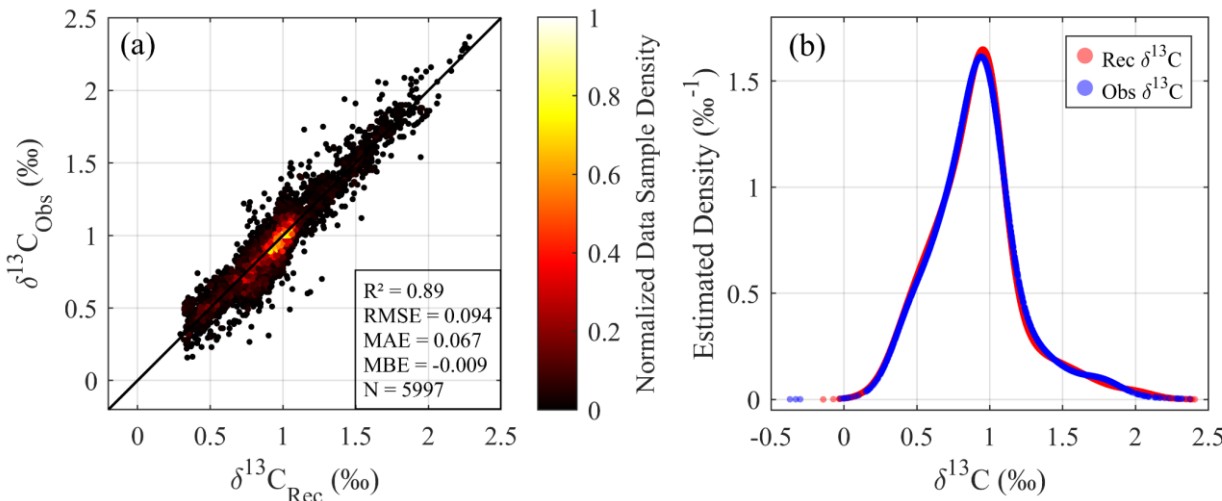

**Figure 5. Comparison of observed and reconstructed $\delta^{13}C_{DIC}$ values derived from the GLODAPv2.2023 dataset.** (a) Density scatter
plot of observed versus reconstructed $\delta^{13}C_{DIC}$ values, using only the intersection of data points with acceptable quality available in both
310    observed and reconstructed datasets (N = 5997). (b) Gaussian kernel density estimations (KDEs) for a comprehensive evaluation of observed
and reconstructed $\delta^{13}C_{DIC}$ values. In panel b, the blue curve represents all acceptable $\delta^{13}C_{DIC}$ observations (N=8941), while the red curve
indicates all acceptable reconstructed $\delta^{13}C_{DIC}$ values (N=68,435), generated by GPR model trained on the GLODAPv2.2023 dataset. The
KDEs reveal the overall distribution shape, highlight differences in peak height and spread, and provide a smooth, continuous representation
of probability density.

315    The GLODAPv2.2023 Atlantic Ocean dataset includes approximately 20,583 stations in total (**Fig. 6a**). Among these, only

about 732 stations have $\delta^{13}C_{DIC}$ observations (blue points in **Fig. 6b**), reflecting the limited spatial and temporal extent of direct

observations due to the high cost and complexity of $\delta^{13}C$ sampling and analysis. Using the proposed GPR model, $\delta^{13}C_{DIC}$

values have been reconstructed at roughly 4,182 stations (red points in **Fig. 6b**). This reconstructed dataset significantly

expands both temporal and spatial coverage of $\delta^{13}C_{DIC}$ in the Atlantic Ocean, enabling more detailed and extensive

320    biogeochemical analyses across the Atlantic.


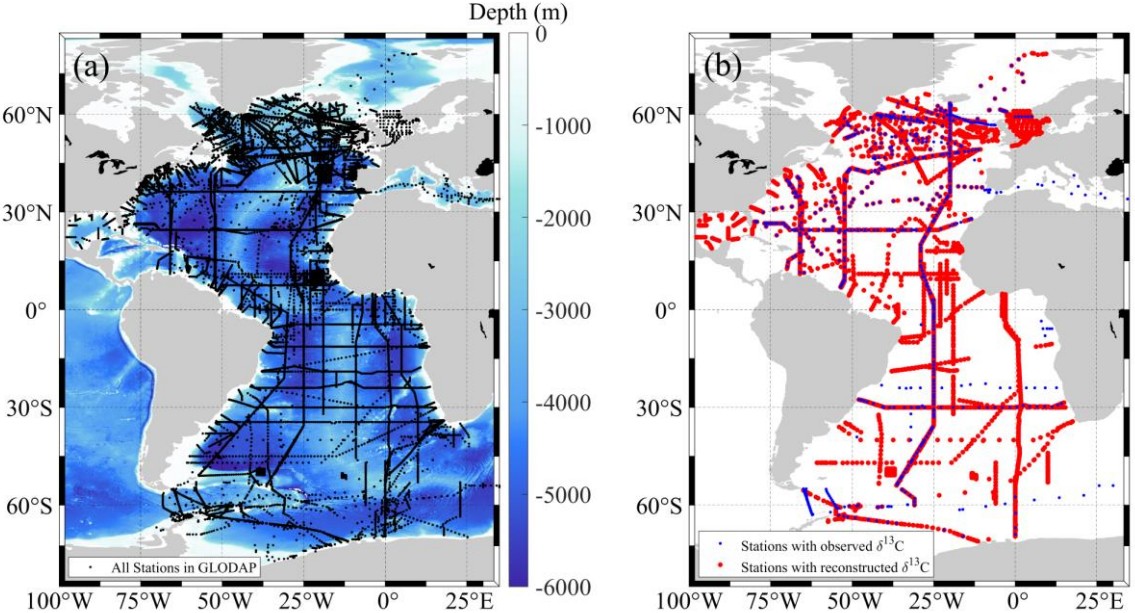

**Figure 6. Station maps for the GLODAPv2.2023 Atlantic Ocean dataset. (a) Locations of all stations included in the dataset. (b) Stations with observed (blue dots) and reconstructed (red dots) $\delta^{13}C_{DIC}$.**

To evaluate the spatial and temporal coverage expansion of the reconstructed $\delta^{13}C_{DIC}$ dataset, the data distributions are compared along longitude, latitude, year, and depth (**Fig. 7**). Along longitude (**Fig. 7a**), the number of reconstructed $\delta^{13}C_{DIC}$ increased several times compared to the observed, and also extended into west of 70° W and east of 0° E where there are almost no direct $\delta^{13}C_{DIC}$ observations. The latitudinal distribution (**Fig. 7b**) shows notable improvements in both hemispheres. The reconstructed dataset greatly enhanced the temporal coverage (**Fig. 7c**). There is little to no reconstructed data before 1980, between 1984 and 1987, and in 1995. The number of reconstructed data increases significantly after the late-1980s, reaching a peak in 2003, (**Fig 7c**). Vertically, the data number of reconstructed $\delta^{13}C_{DIC}$ increased substantially throughout the water column, although numbers of both reconstructed and observed $\delta^{13}C_{DIC}$ data show exponential decrease from surface to deep water (**Fig. 7d**).

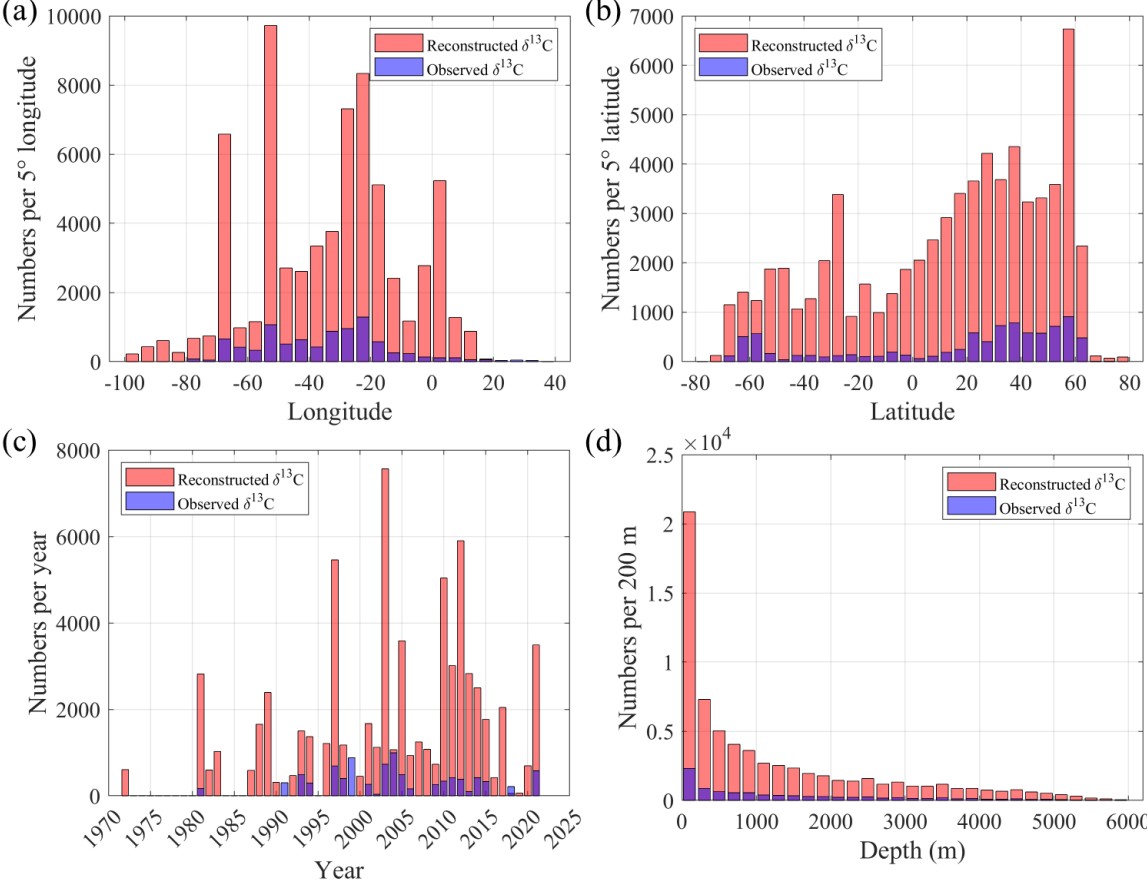

**Figure 7. Comparison of the spatial, temporal, and vertical distributions of observed and reconstructed δ¹³C_DIC data in the Atlantic Ocean.** The number of δ¹³C_DIC samples is shown by (a) longitude (per 5° bin), (b) latitude (per 5° bin), (c) year (from 1972 to 2021), and (d) depth (per 200 m interval). Red bars represent reconstructed δ¹³C_DIC, and blue bars indicate observed δ¹³C_DIC.

Overall, the reconstructed δ¹³C_DIC dataset consistently demonstrated good accuracy, high correlation coefficient, low RMSE, MAE, and minimal bias across diverse environmental conditions. Spatially, the reconstruction significantly expands coverage across longitude, latitude, and depth, particularly in undersampled regions such as the South Atlantic and deeper ocean. Temporally, it enhances data availability, filling many before 1990 and after 2015, despite some limitations tied to input variable availability.





## 3.5 Potential Implications

The improved spatial, temporal, and vertical coverage of the reconstructed $\delta^{13}C_{DIC}$ dataset potentially contributes to the biogeochemical research and to a deeper understanding of carbon cycling processes in the Atlantic Ocean. Two specific
applications are given as examples here: (1) examining the spatial patterns and decadal variability of surface $\delta^{13}C_{DIC}$ ($\leq 10$ m), and (2) evaluating its ability to resolve vertical profiles, thereby providing insights into subsurface carbon dynamics.

To facilitate a more comprehensive understanding of the spatial and temporal dynamics of surface $\delta^{13}C_{DIC}$ in the Atlantic, **Fig. 8** illustrates how the reconstruction improves the spatial representativeness and temporal continuity of surface $\delta^{13}C_{DIC}$. The expanded spatial coverage shown in **Fig. 8a** increases the representativeness of latitudinal sample bins, reducing the influence
of sparsely sampled outliers on bin means (**Fig. 8b**). For instance, observational data in the 5° S – 10° S band during the 2000s show an anomalously low mean that is likely driven by a few isolated observations. In contrast, the reconstructed dataset, by supplying a larger and spatially coherent sample set, yields a mean that is more consistent with adjacent latitude bands. This effect should be understood as a reduction of sampling-driven discontinuities and noise rather than as an artificial suppression of genuine signals. Both observed and reconstructed data indicate a basin-scale decline in surface $\delta^{13}C_{DIC}$ from the 1980s to
the 2010s, most pronounced in tropical and subtropical regions (approximately 35° S – 35° N), consistent with the expected Suess effect from increasing fossil-fuel $CO_2$ (Keeling, 1979). Mid- to high-latitude regions display more complex, regionally heterogeneous variability, likely reflecting circulation, water-mass formation and subduction processes. In contrast, the mid- to high-latitude regions exhibit more complex variability, likely reflecting the influence of regional circulation, water mass formation, and subduction processes that modulate isotopic signals on multi-decadal timescales.

To facilitate a clear and intuitive assessment of decadal distributions of observed and reconstructed surface $\delta^{13}C_{DIC}$, we employ KDEs. KDEs presented in **Fig. 8c** are computed using all acceptable observed and reconstructed surface $\delta^{13}C_{DIC}$ samples within each decade and are intended to reveal the overall distributional characteristics and their evolution across decades and complement the latitude-binned means shown in **Fig. 8b**. In general, both observed and reconstructed KDEs confirm the decadal downward shifts of $\delta^{13}C_{DIC}$ from the 1980s to the 2010s, reinforcing the notion of a basin-wide isotopic response to
anthropogenic carbon inputs. And the observed and reconstructed KDEs within each decade fall within comparable value ranges and exhibit similar peak positions, underscoring the consistency between the two datasets. Notable discrepancies arise in specific decades. For instance, in the 1980s the reconstructed KDE shows a dominant peak near 2 ‰ and a secondary peak near 1.7 ‰. The latter closely corresponds to the single, broader maximum in the observational KDE. This discrepancy is likely attributable to the sparse observational coverage during this decade (**Fig. 8a**), which limits the representativeness of the
latitudinal distribution. In the 2000s, the reconstructed KDE exhibits slightly greater density at the lower end of the common range (~0.5–2.0‰) than the observational KDE, producing a modest negative shift in central tendency while maintaining a comparable overall span. This shift is primarily attributable to a disproportionate increase in reconstructed data between 50°

N and 65° N, which modifies the density structure of the distribution. These examples highlight that the reconstruction not only reproduces the principal features of the observational distributions but also mitigates the limitations imposed by sparse or uneven sampling. The reconstructed KDEs are generally smoother and more spatially coherent than the raw observational KDEs, which reduces sampling-induced noise and preserves large-scale signals. As a result, the reconstruction provides a more representative depiction of $\delta^{13}C_{DIC}$ variability across latitude and time, improving the statistical robustness and interpretability of the inferred distributional changes.

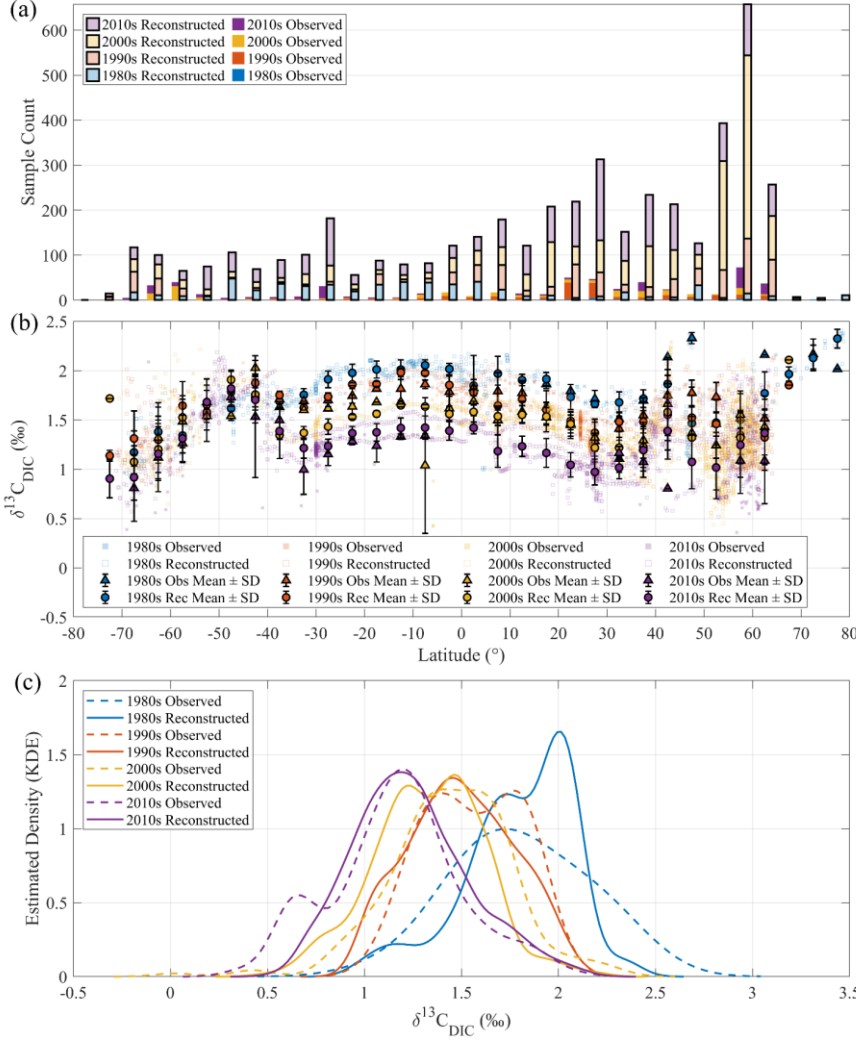

**Figure 8. Surface distribution of observed and reconstructed $\delta^{13}C_{DIC}$ in the Atlantic Ocean across four decades.** (a) Number of observed and reconstructed $\delta^{13}C_{DIC}$ samples within each 5° latitude band for the 1980s, 1990s, 2000s, and 2010s. For each latitude band,



stacked bars represent decadal sample counts. Solid (opaque) bars indicate observations, and semi-transparent bars with black outlines indicate corresponding reconstructions. (b) Surface $\delta^{13}C_{DIC}$ distributions (depth $\leqslant 10$ m) along latitude for the 1980s, 1990s, 2000s, and 2010s. Colored scatter points indicate individual $\delta^{13}C_{DIC}$ values from observations (circles) and reconstructions (dots), while the black

markers with corresponding face color and error bars represent the mean ± standard deviation of $\delta^{13}C_{DIC}$ within each 5° latitude bin, triangles for observed data and diamonds for reconstructed values. (c) Gaussian kernel density estimates (KDE) of surface $\delta^{13}C_{DIC}$ for each decade (1980s–2010s), using dashed lines for observations and solid lines for reconstructions. These KDEs reveal the decadal evolution and overall distributional characteristics of $\delta^{13}C_{DIC}$ in both datasets.

Besides horizontal distributions, the reconstructed $\delta^{13}C_{DIC}$ dataset also provides valuable insights into vertical variability. The

depth profiles along the North Atlantic A16N section in 1993, 2003, 2013, and 2023 (**Fig. 9**) show that the reconstruction substantially improves vertical resolution and continuity, especially for years with sparse measurements. For instance, the $\delta^{13}C_{DIC}$ samples were increased from 526 to 2,199 in 1993, 38 to 1,500 in 2003, and 498 to 2,504 in 2013, respectively, enhancing data coverage across depths and latitudes, facilitating the detection of temporal trends associated with ocean carbon uptake and redistribution (**Fig. 9**). Specifically, the 2023 reconstruction was generated using the observational predictors

collected along A16N in 2023 (available from CCHDO). For 2023 with a high-density observational dataset, the reconstructed data closely align with the observed data (**Fig. 9d vs. 9h**).

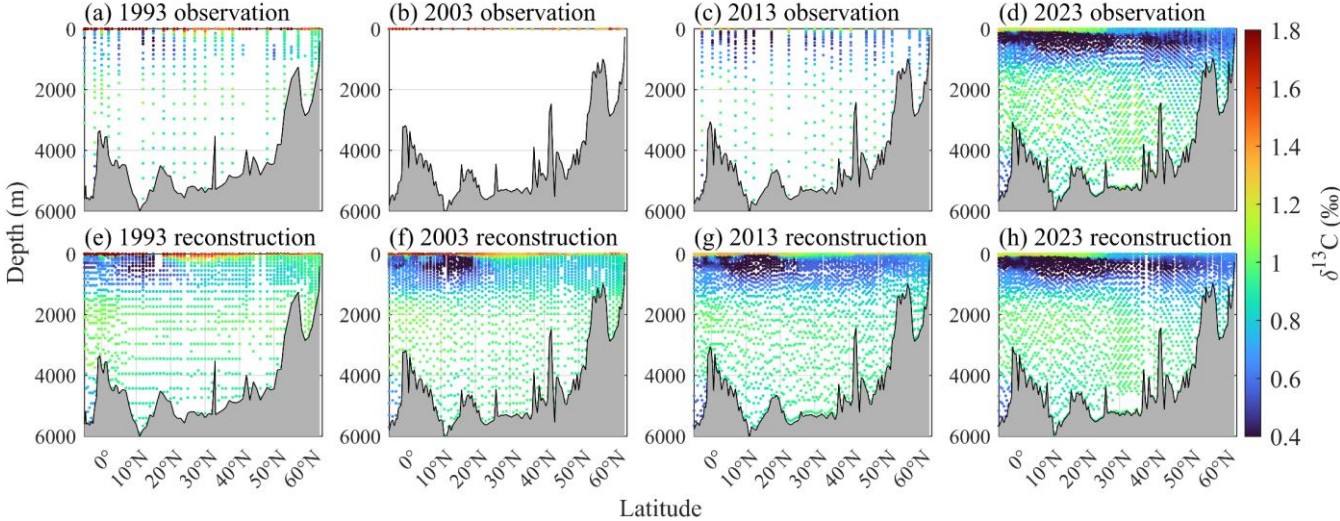

**Figure 9. Enhancement of spatial resolution for $\delta^{13}C_{DIC}$ through GPR model reconstruction along North Atlantic section A16N.** Top panels (a–d) presents in-situ observed $\delta^{13}C_{DIC}$ from cruises in (a) 1993 (n = 526), (b) 2003 (surface-only, n = 38), (c) 2013 (n = 498), and (d)

2023 (n = 3,460). Bottom panels (e–h) shows corresponding reconstructed $\delta^{13}C_{DIC}$ with systematic filling of spatial gaps: (e) the number of reconstructed $\delta^{13}C_{DIC}$ for cruise A16N in 1993 is 1,500; (f) full-depth reconstruction for cruise A16N in 2003 (n = 2,199); (g) 2013 reconstruction (n = 2,504), highlighting improved coverage compared to limited observations (n = 498); (h) the reconstruction of cruise A16N in 2023 validated against high-density observations. Reconstruction effectively resolves historical sampling sparsity, yielding consistent spatial resolution comparable to contemporary DIC measurements.



The enhanced vertical profiles provided by the reconstructed $\delta^{13}C_{DIC}$ dataset also enable more refined investigations into the physical and biological controls governing carbon isotope variability in the ocean interior. Joint interpretation of $\delta^{13}C_{DIC}$ with nutrients such as phosphate allows for the identification of water mass structures and biological removal and addition, shedding light on processes such as organic carbon export, remineralization, and the coupling between biological productivity and ocean circulation (e.g., Gruber et al., 1999; Eide et al., 2017). These capabilities improve the isolation of the vertical imprint of the

Suess effect and facilitate the reconstruction of preindustrial $\delta^{13}C_{DIC}$, both of which are critical for assessing anthropogenic perturbations to the marine carbon cycle (Olsen et al., 2010). $\delta^{13}C_{DIC}$ also can be used to estimate biological carbon export and net community production (NCP) (Quay et al., 2009; Yang et al., 2019). Accurate NCP estimation from $\delta^{13}C_{DIC}$ requires knowledge of both the physical supply of DIC and $\delta^{13}C_{DIC}$, typically represented by the subsurface to surface $\delta^{13}C_{DIC}$ gradient. Sparse historical $\delta^{13}C_{DIC}$ data can bias this estimate, whereas the reconstructed dataset provides a continuous field that reduces

gaps and noise, enabling more reliable NCP calculations. Importantly, the observation-constrained reconstructed $\delta^{13}C_{DIC}$ fields fill longstanding gaps in global datasets, providing a robust basis for the validation of Earth system models (e.g., Schmittner et al., 2013; Sonnerup & Quay, 2012; Claret et al., 2021). The improved coverage helps reconcile discrepancies between modeled and observed $\delta^{13}C_{DIC}$ distributions, particularly in data-sparse regions such as the deep ocean and the Southern Hemisphere. In particular, at the boundary of two water masses, the high resolution $\delta^{13}C_{DIC}$ distribution helps to validate high-

resolution global physical and biogeochemical model predictions and more effectively study carbon cycle at such boundary zones.

Overall, the reconstruction effectively addresses key limitations of the observational $\delta^{13}C_{DIC}$ record, particularly data sparsity and sampling bias, providing a more continuous and spatially balanced dataset. This enables clearer identification of large-scale latitudinal gradients, decadal trends, and regional anomalies, offering a robust foundation for interpreting long-term

carbon cycle dynamics.

### 3.6 Challenges and Limitations

Despite enhancing spatial resolution, reconstructing grid $\delta^{13}C_{DIC}$ with low uncertainty remains challenging compared to other carbonate system variables (e.g., DIC, $p$CO$_2$, TA), primarily due to limited historical observation coverage. While the dataset achieves an overall mean bias of −0.007 ‰, notable regional discrepancies persist, particularly in subpolar regions, where

biases of some samples exceed 0.15 ‰. These anomalies may be attributed to nonlinear interactions between $\delta^{13}C_{DIC}$ and biogeochemical processes, such as upwelling-induced isotope fractionation and biological carbon pump effects (Gruber et al., 1999). Additionally, although input variables (T, S, AOU, N, Si, DIC, $x$CO$_2$) were validated to have negligible impacts on reconstruction uncertainty ($3.77 \times 10^{-14}$ ‰), the exclusion of potential covariates, such as chlorophyll-a, wind speed, or nutrient gradients, may limit constraint precision.

The reconstruction approach is also subject to inherent limitations, such as spatial interpolation assumptions, uncertainty propagation, and temporal variability constraints. The GPR model assumes stationarity of $\delta^{13}C_{DIC}$ spatial covariance, a simplification that may fail in regions with abrupt bathymetric changes (e.g., the Mid-Atlantic Ridge). Such nonstationarity may introduce systematic errors in reconstructed $\delta^{13}C_{DIC}$ gradients, particularly across topographic features that influence ocean circulation and carbon transport. Cumulative uncertainties from measurement (0.07 ‰), mapping (0.08 ‰), and input

variables ($3.77 \times 10^{-14}$ ‰) yield an overall uncertainty of 0.11 ‰, which may obscure small-scale $\delta^{13}C_{DIC}$ signals. This limitation restricts the dataset's utility for resolving fine-scale temporal trends or localized isotopic anomalies. Furthermore, the temporal sparsity of historical predictor observations restricts the resolution of interannual $\delta^{13}C_{DIC}$ variability. The application of reconstructed data to estimate the Suess effect using extended Multilinear Regression (eMLR) methods in repeated hydrographic transects requires careful consideration due to inherent uncertainties. The decadal-scale Suess effect,

approximately 0.2 ‰, is often comparable to parts of discrepancy between reconstructed and observed $\delta^{13}C_{DIC}$ values. Consequently, rigorous assessment of these uncertainties is critical to ensure the reliability of decadal-scale isotopic trend estimates derived from eMLR analyses.

To address these challenges and limitations, future improvements will aim at enhancing the mechanistic understanding of relationships between $\delta^{13}C_{DIC}$ and environment factors by developing subregional partitioning strategies and variable selection

frameworks that optimize model fidelity, integrating process-based insights to identify optimal predictor variables and refine regionalization schemes for biogeochemical heterogeneity. Additionally, high-resolution wind speed data may be incorporated to resolve air-sea gas exchange effects, coupled with uncertainty mitigation techniques, while emerging high-resolution observations from autonomous platforms may be leveraged to refine spatial resolution to fixed grids. Furthermore, multi-phase reconstruction frameworks have the potential to extend $\delta^{13}C_{DIC}$ records across pre- and post-satellite eras using proxy variables

for historical biological productivity, improving temporal consistency and Suess effect estimation via eMLR methods by reducing uncertainties in decadal-scale trend analyses.

## 4 Conclusion

This study reconstructs Atlantic Ocean $\delta^{13}C_{DIC}$ using a GPR model trained on 37 secondary quality-controlled cruises selected from 51 compiled historical datasets, including a high-resolution 2023 A16N section. GPR was chosen for its ability to capture

nonlinear relationships, outperforming other machine-learning models in preliminary tests (lowest RMSE and highest $R^2$). The trained GPR model achieved an average bias of $-0.007 \pm 0.082$ ‰ and an overall uncertainty of 0.11 ‰, with error contributions from measurement (0.07 ‰), mapping (0.08 ‰) and input-variable ($3.77 \times 10^{-14}$ ‰).

Applying GLODAPv2.2023 Atlantic dataset as predictors, the reconstruction expanded the acceptable $\delta^{13}C_{DIC}$ samples from 8,941 to 68,435, representing a substantial 7.65-fold increase compared to the original dataset. This extensive dataset significantly improves spatial coverage in longitude, latitude, and depth and enhances temporal continuity of $\delta^{13}C_{DIC}$ observations over the past four decades.

Multiple evidences attest to reliability and superiority of the reconstructed dataset. Statistical diagnostics demonstrate strong model skill in both cross-validation and independent testing stages, as well as reconstruction stage. The reconstruction preserves decadal variability at the sea surface $\delta^{13}C_{DIC}$ and in depth-profiles and agrees with high-density contemporary observations such as the 2023 A16N section. Distributional metrics, exemplified by smoothed and stable KDEs, suggest that the reconstruction mitigates sampling noise while retaining meaningful spatiotemporal signals.

The broad spatial coherence and high coverage of the reconstructed $\delta^{13}C_{DIC}$ enable systematic analysis of large-scale gradients, detection of regional anomalies, and investigation of tracer relationships with nutrients such as phosphate, supporting a wide range of carbon cycle studies. It also provides a valuable baseline for evaluation of Earth system models, for improving estimates of preindustrial and anthropogenic $\delta^{13}C_{DIC}$, and for extending isotopic records used in climate reanalysis.

Despite these advances, some challenges and limitations remain. Regional biases persist, likely reflecting nonlinear biogeochemical interactions and the GPR assumption of spatial covariance stationarity, which may break down over complex bathymetry. Small-scale features can be obscured by cumulative uncertainty propagation and temporal inconsistencies persist in certain derived quantities. Future efforts will be needed on assimilating higher-resolution observations to enhancing the mechanistic understanding of relationships between $\delta^{13}C_{DIC}$ and environment factors, developing tailored subregional modelling frameworks and exploiting advanced machine-learning techniques to capture nonstationary spatial features. These efforts will refine spatial and temporal fidelity, reduce uncertainties and gain deeper mechanistic insight into ocean carbon cycle dynamics.

## 5 Data availability

The qualified-controlled and reconstructed $\delta^{13}C_{DIC}$ in the Atlantic Ocean are available as two NetCDF/Excel files at https://doi.org/10.5281/zenodo.16907402

## 6 Code availability

The MATLAB code used to process the data and create the figures included in this paper is provided at https://github.com/huigao109/ReC13_ML

## Author contribution

**Hui Gao**: Conceptualization, data curation, formal analysis, methodology, software, visualization, writing – original draft preparation, writing – review & editing. **Zelun Wu**: Validation, methodology, writing – review & editing. **Zhentao Sun**: Validation, writing – review & editing. **Diana Cai**: Validation, methodology, writing – review & editing. **Meibing Jin**: Conceptualization, supervision, validation, writing – review & editing. **Wei-Jun Cai**: Conceptualization, funding acquisition, methodology, validation, writing – review & editing, project administration, supervision.

## Competing interests

The authors declare that they have no conflict of interest.

## Acknowledgments

The authors gratefully acknowledge all researchers, principal investigators, captains, and crew members who contributed their time at sea and in the laboratory to collect, analyze, and curate the datasets used in this study. We thank the providers and maintainers of the GLODAP, OCADS, and CCHDO databases, as well as the creators of the internally consistent $\delta^{13}C_{DIC}$ dataset for the North Atlantic Ocean (NAC13v1; Becker et al., 2016), for making their data openly available. The authors used ChatGPT for grammar and language polishing; all scientific content, interpretations, and conclusions are the sole responsibility of the authors.

## Financial support

This research is supported by the US National Science Foundation awards (OCE-2123768 & OCE-25A00158) to W-J.C.

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
