# Peer review of "Reconstruction of $\delta^{13}C_{DIC}$ in the Atlantic Ocean: A Probabilistic Machine Learning Approach for Filling Historical Data Gaps"

_Earth System Science Data, 2025_

## Author Comment (AC1)

We are grateful to Dr. Patrick Rafter for dedicating his time and providing constructive comments, which are instrumental in refining this manuscript. Below, we have thoroughly addressed every comment, and the original review text is presented in italics.

*The manuscript "Reconstruction of d13CDIC in the Atlantic Ocean…" as reviewed by Patrick Rafter*

*First, I'd like to thank the other (anonymous) reviewer for their careful and useful review of this manuscript. If I were the author of this manuscript, I would greatly appreciate the many meaningful and well-informed comments. I don't fully agree with all their suggestions, but it is undeniably a high-quality review.*

*For example, I think—for the most part—this study needs less additional work than the other reviewer. The suggestion to implement the ML method in a model environment would be a very interesting and valuable addition to this work, but I predict the authors' response will be "outside the scope of the current study". It sounds to me like a huge amount of new work, but I may be incorrect in this (or it may just be a huge amount of work for \*me\* and not someone else (it almost surely is)). Note that I do not have the experience in this space to comment on whether this model environment application is "now common practice", but I will say that this would have been a novel (to me), interesting, and seemingly robust application of the methods developed here. But I would like to note that if this manuscript / dataset were to follow the reviewer's advice, it would boost my score for the "significance" and "data quality" categories into and above the 'Excellent' category. As of now, I have scored these as 'good'.*

*I also think the motivation is appropriate for this specific study and that the decadal trends in the Kernel Density Estimates (see Fig. 8) are an interesting outcome from this study (as it exists now).*

R: Thank you for your positive and constructive feedback on our manuscript. We greatly appreciate your recognition of the study's value and your thoughtful reflections on the other reviewer's comments.

Regarding the suggestion to validate the Machine Learning method in a model environment (raised by the other reviewer), we are pleased to inform you that we implemented this supplementary validation, and it was indeed feasible within the scope of the current study. As noted in our response to the other reviewer, the model dataset they referenced (https://doi.org/10.5194/gmd-17-1709-2024) does not provide the required carbon isotope data. Instead, we adopted the well-validated model data from Claret et al. (2021), which includes comprehensive carbon isotope simulations ideal for this validation purpose. Following the proposed workflow, we subsampled the model outputs across time and space, reconstructed the 4D $\delta^{13}C_{DIC}$ distribution, and thoroughly evaluated the model's performance.

This supplementary validation not only confirms the method's ability to accurately reconstruct spatiotemporal patterns from sparse and noisy data but also reveals its strengths in mitigating sampling biases, effectively addressing the limitations of validating solely with sparse observations. All details of this model-based validation, including data processing steps, evaluation metrics, and key results, have been added to the Appendix of the revised manuscript for transparency and reference.

We are grateful for your note that this additional work would enhance the study's "significance"

and "data quality" categories. By incorporating this model-based validation, we aim to strengthen the scientific rigor and reliability of our research as you suggested. Thank you again for your valuable input and support. Your feedback has been instrumental in refining our work.

Claret, M., Sonnerup, R. E., and Quay, P. D.: A Next Generation Ocean Carbon Isotope Model for Climate Studies I: Steady State Controls on Ocean 13 C, Global Biogeochemical Cycles, 35, e2020GB006757, https://doi.org/10.1029/2020GB006757, 2021.

*Where I agree with the anonymous reviewer is that I think the new "reconstructed" dataset could be (I think): (1) expanded spatially using the GLODAP gridded product and (2) that this would be a very useful addition to our community. I am assuming these are "minor revisions" as the ML model is already built and I assume the application to the gridded product will be straightforward (and worth the time for the community to use!). I would also urge the authors to consider the other options listed by the anonymous reviewer to expand the ML methods temporally, although I am unfamiliar with the reviewer's specific suggestions and cannot comment on the time requirements for such new applications.*

*Likewise, the other reviewer makes strong comments about the dataset itself. I agree that adding the reconstructed dataset as its own column (with -999 for other basins) to the existing GLODAP data would be very useful for the community. Even better would be for the community to have a gridded product!*

R: Thank you for your valuable feedback and recognition of the community utility of our reconstructed dataset. We fully agree with your suggestions regarding spatial expansion and dataset compatibility.

Regarding your suggestion to expand spatially using the GLODAP gridded product, we have thoroughly checked the official GLODAP repository but have not found an official gridded version of the dataset. We fully acknowledge the value of a gridded $\delta^{13}C_{DIC}$ product for the community and would be pleased to supplement our reconstruction with a corresponding gridded product if an official GLODAP gridded product becomes available. However, given the current spatiotemporal sparsity of the underlying $\delta^{13}C_{DIC}$ observations, we cautiously note that direct gridding at this stage may introduce additional uncertainties, including over interpolation in data-sparse regions and potential misrepresentation of true biogeochemical variability. This is a key consideration for maintaining the scientific rigor of the product, as our priority is to provide a reliable dataset that reflects the actual constraints of available observations.

We also sincerely appreciate your suggestion to enhance compatibility with GLODAP. However, as GLODAPv2 provides separate, official datasets for individual ocean basins (e.g., Atlantic, Pacific, Indian, Arctic), we have retained our product's focus on the Atlantic Ocean, which is consistent with this basin-specific framework, rather than adding the reconstructed $\delta^{13}C_{DIC}$ to the full global GLODAPv2.2023 dataset (with non-Atlantic basins set to -999). This approach avoids unnecessary redundancy, as users can already access GLODAP's global or other basin datasets directly from the official repository and seamlessly merge them with our Atlantic product as needed. To ensure clarity and interoperability, we have updated the Zenodo archive with a detailed README file. This document explains the dataset structure, labels for new fields, and step-by-step guidance for merging our product with GLODAPv2's global or basin-specific datasets. We have also clarified citation requirements in both the manuscript and Zenodo metadata, emphasizing that

users should cite GLODAPv2 for native variables and our work for the reconstructed $\delta^{13}C_{DIC}$.

Regarding your suggestion to expand the Machine Learning method temporally, we fully recognize the value of a spatiotemporally continuous $\delta^{13}C_{DIC}$ product and view this as a critical next step. As noted in our response to the anonymous reviewer, the current study prioritizes addressing spatial sparsity, an urgent gap given the extreme paucity of $\delta^{13}C_{DIC}$ observations. Temporal extension requires robust constraints on seasonal/interannual variability, which are currently limited by uneven temporal coverage of existing observations (most concentrated in summer). To advance this, we plan to integrate long-term time-series data from programs like BIOS (Bermuda) and HOT (Hawaii) to calibrate the ML model for temporal dynamics, building on the validated spatial reconstruction framework. We will also more fully use the numerical model data to validate the future work.

Again, thank you for your constructive suggestions. Your suggestions have helped us refine the utility and transparency of our dataset, and we remain committed to enhancing its value for the community in future work.

*Below I have listed notes I made on the manuscript as I read through it.*
*Line by line notes*
*27: need to define delta notation*

R: We appreciate your comment pointing out the need to define delta notation. In the revised manuscript, we have supplemented this definition when $\delta^{13}C$ is first introduced: "The stable carbon isotope ratio, $\delta^{13}C$ (expressed via the standard delta notation: $\delta^{13}C=(({}^{13}C/{}^{12}C)_{sample}/({}^{13}C/{}^{12}C)_{standard}-1)\times10^{3}$, with the international reference standard usually the Vienna Pee Dee Belemnite ([V]-PDB) fossil), has been widely applied as a tracer in marine carbon research, providing valuable insights into various processes within the oceanic carbon system."

*79+: I don't see a need to shorten "Section" here*

R: Thanks for your comment. We agree with your view that there is no need to abbreviate "Section" here. To align with your suggestion and enhance readability, we have revised the original text by restoring all abbreviated "Sect." to the full term "Section".

*100: I like the previous paragraph*

R: Thanks for your comment.

*132: what exactly does "exhibit high internal consistency" mean? Are there statistics to support this statement?*

R: Thanks for your comment. To clarify, this phrase aligns with the definition used in Becker et al. (2016) and refers to the quantifiable agreement between overlapping data points within the dataset, ensuring no contradictory or anomalous deviations that would compromise reliability. Its core lies in verifying the consistency of data within the dataset through quantitative calculations. The specific explanation and statistical support are as follows.

Here, "high internal consistency" refers to a high level of coordination and reliability among the various data points within the final dataset, with no significant contradictions or abnormal deviations. This consistency is not a subjective judgment but a conclusion drawn from quantitative calculations of the "offsets" at "crossovers" in the dataset, ensuring the dataset has logical stability

internally.

The statistical calculation for this conclusion refers to the method proposed by Tanhua et al., 2010, with specific steps as follows: The "Weighted Mean (WM)" is used to quantify the internal consistency of the dataset. The weight is determined by the offset of each crossover and its standard deviation, which emphasizes the influence of more reliable data on the result.

$$WM = \frac{\sum_{i=1}^{L} D(i)/(\sigma(i))^2}{\sum_{i=1}^{L} 1/(\sigma(i))^2}$$

Parameter Definitions:

L: Represents the total number of crossovers in the dataset.

D(i): Refers to the respective offset of the i-th crossover (i.e., the numerical difference of different data at that crossover).

σ(i): Denotes the standard deviation of the offset of the i-th crossover, which is used to measure the degree of dispersion and reliability of that offset.

Becker, M., Andersen, N., Erlenkeuser, H., Humphreys, M. P., Tanhua, T., and Körtzinger, A.: An internally consistent dataset of δ13C-DIC in the North Atlantic Ocean – NAC13v1, Earth System Science Data 8: 559-570, https://doi.org/10.5194/essd-8-559-2016, 2016.

Tanhua, T., van Heuven, S., Key, R. M., Velo, A., Olsen, A., and Schirnick, C.: Quality control procedures and methods of the CARINA database, Earth Syst. Sci. Data 2: 35-49, https://doi.org/10.5194/essd-2-35-2010, 2010.

*139: Is GPR an acronym? Perhaps not relevant, but I wanted to know*

R: Thank you for your question. GPR is the acronym for Gaussian Process Regression, which first appears in the Introduction (Line 74).

*161: Repeated text*

R: Thanks for your comment. We have deleted these sentences and reorganized the sentences in Section 2.2 (Line 133-135) as: "After applying additional adjustments, the $\delta^{13}C_{DIC}$ data for the remaining 37 cruises exhibit high internal consistency. These 37 cruises do not include 13 cruises without deep-water crossover stations (Table 1) and cruise 64TR19900417, which were excluded to ensure data reliability as their uncertainties cannot be objectively quantified. Collectively, these excluded cruises accounted for less than 3 % of total $\delta^{13}C_{DIC}$ measurements."

*Fig. 2: I like the figure, but as the other reviewer noted, it would be better to use completely independent cruise datasets for the validation as well as the "independent" tests*

R: Thank you for your feedback. We would like to clarify that our existing validation framework design, which aligns with this core principle while balancing statistical robustness and practical feasibility for sparse oceanographic data.

As detailed in our response to the first reviewer, our independent test set was intentionally selected to be fully decoupled from the training/validation pool, with no overlap in cruises, spatial regions, or temporal coverage. This means measurements from any cruise are entirely confined to either the training/validation set or the independent test set, ensuring the final performance evaluation (reported RMSE and R²) is based on completely unseen cruises, which directly addresses the need for independent cruise-based testing. The 10-fold cross-validation within the training set

was solely for hyperparameter tuning, not for final performance assessment, so it does not compromise the independence of the test phase.

We maintained this design because many of the 51 cruises in our dataset have small sample sizes. Splitting the training/validation set by cruise would result in highly imbalanced folds, leading to unstable hyperparameter tuning and biased cross-validation results, undermining the statistical rigor of the model development process. By using random splitting within the training/validation pool, we preserve the natural spatiotemporal variability of $\delta^{13}C_{DIC}$ data, ensuring the model is tuned to generalize across diverse oceanic conditions rather than specific cruises. This approach is consistent with established practices in oceanographic ML studies (e.g., Lima et al., 2023; Regier et al., 2023; Wu et al., 2025), as cited in our response to the first reviewer.

To enhance clarity on the independence of the cruise datasets, we have revised the relevant paragraph to explicitly highlight that the independent test set comprises completely separate cruises from the training/validation pool: "The dataset was randomly split into a training set (80%) and a validation set (20%), with model training and hyperparameter tuning performed using 10-fold cross-validation within the training set to mitigate overfitting. An independent test set was reserved for final performance evaluation, selected to ensure no overlap with the training/validation set in cruises, spatial regions, or temporal coverage. We opted for random splitting over cruise-separated k-fold cross-validation to balance robustness and feasibility: many of the 51 cruises have small sample sizes, and cruise-separated splitting would cause imbalanced folds, leading to unstable hyperparameter tuning and biased results. Random splitting also preserves the natural spatiotemporal variability of $\delta^{13}C_{DIC}$, tuning the model to generalize across diverse oceanic conditions rather than specific cruises. This framework aligns with established practices for sparse oceanographic datasets (Lima et al., 2023; Regier et al., 2023; Wu et al., 2025).". This revision ensures the manuscript clearly conveys that our validation strategy incorporates fully independent cruise datasets for the critical final evaluation, while the training-phase cross-validation design prioritizes practical feasibility and stable model tuning.

Lima, I. D., Wang, Z. A., Cameron, L. P., Grabowski, J. H., & Rheuban, J. E.: Predicting Carbonate Chemistry on the Northwest Atlantic Shelf Using Neural Networks. Journal of Geophysical Research: Biogeosciences, 128(7), e2023JG007536. https://doi.org/10.1029/2023JG007536, 2023.

Regier, P., Duggan, M., Myers-Pigg, A., & Ward, N.: Effects of random forest modeling decisions on biogeochemical time series predictions. Limnology and Oceanography: Methods, 21(1), 40-52, https://doi.org/10.1002/lom3.10523, 2023.

Wu, Z., Lu, W., Roobaert, A., Song, L., Yan, X.-H., and Cai, W.-J.: A machine-learning reconstruction of sea surface p CO2 in the North American Atlantic Coastal Ocean Margin from 1993 to 2021, Earth Syst. Sci. Data, 17, 43–63, https://doi.org/10.5194/essd-17-43-2025, 2025.

*192: I wonder if other Earth scientists would be as surprised to learn of Mean Absolute Error and Mean Bias Error. I think they might and it might therefore be useful to use a sentence or two describing why these additional metrics are useful to the study*

R: Thank you for your suggestion. As recommended, we have supplemented the original sentence about model accuracy evaluation with 2-3 sentences (Line 205-209: "Among these metrics, MAE and MBE are valuable for evaluating the performance of the machine learning models. MAE

quantifies the average absolute deviation between observed and predicted values; its insensitivity to outliers makes it ideal for handling the potential noise in $\delta^{13}C_{DIC}$ observational data, ensuring a robust measure of overall prediction error. MBE, by retaining the sign of deviations, identifies systematic biases (e.g., consistent overestimation or underestimation of $\delta^{13}C_{DIC}$), which is critical for refining the machine learning model.") explaining MAE and MBE, with a specific focus on their application in machine learning, to enhance the manuscript's clarity for the broader community.

*202: Propagated error?*

R: Yes, the total uncertainty of the reconstructed $\delta^{13}C_{DIC}$ is a propagated error. As detailed in the revised manuscript, we assumed independence between the three uncertainty sources ($u_{obs}$, $u_{inputs}$, $u_{map}$) and calculated the total uncertainty using the standard error propagation method (root-sum-of-squares synthesis), as supported by the cited references (Hughes and Hase, 2010; Taylor, 1997). We have refined the text to explicitly emphasize the error propagation approach and its implementation, ensuring clarity on this point.

Revised Text in Manuscript:

The comprehensive uncertainty of the reconstructed $\delta^{13}C_{DIC}$ was derived via error propagation, assuming independence between distinct uncertainty sources. These sources of uncertainties include: the direct $\delta^{13}C_{DIC}$ measurement uncertainty from observations ($u_{obs}$), the uncertainty accumulated from the input variables ($u_{inputs}$), and the uncertainty induced by the mapping function ($u_{map}$). Following standard error propagation protocols (Hughes and Hase, 2010; Taylor, 1997), the comprehensive uncertainty of our estimated $\delta^{13}C_{DIC}$ product, $u_{\delta^{13}C_{DIC}}$, was calculated as the root sum of the squares of the individual uncertainties:

$$u_{\delta^{13}C_{DIC}} = \sqrt{u_{obs}^2 + u_{inputs}^2 + u_{map}^2}$$

*212: perturbed not perturbs*

R: We are grateful for you noticing this typo. The error has been fixed in the corresponding section of the revised manuscript.

*230: I'm unsure where the 10-fold cross-validation comes from*

R: 10-fold cross-validation was selected based on its availability as a standard option in MATLAB's Machine Learning Toolbox, which is widely used for model training in our field (e.g., Wu et al., 2025), and its suitability for balancing computational efficiency and generalization performance with our dataset. We have added this clarification to the manuscript to enhance transparency (Line ): "During the training phase, we leveraged a 10-fold cross-validation approach, selected as it is a standard pre-implemented option in MATLAB's Machine Learning Toolbox. This approach balances computational efficiency and robustness, reducing overfitting by iteratively splitting training data into 10 folds: 9 for training and 1 for validation per iteration, with results averaged across iterations to ensure stable performance. Finally, the model achieved an $R^2$ of 0.92, an RMSE of 0.083 ‰, an MAE of 0.056 ‰, and an MBE of −0.0003 ‰ (**Fig. 3a**)."

Wu, Z., Lu, W., Roobaert, A., Song, L., Yan, X.-H., and Cai, W.-J.: A machine-learning reconstruction of sea surface p CO2 in the North American Atlantic Coastal Ocean Margin from 1993 to 2021, Earth Syst. Sci. Data, 17, 43–63, https://doi.org/10.5194/essd-17-43-2025, 2025.

*249: This text is also somewhat a repetition of earlier text*

R: Thanks for your comment. We revised this sentence as: "To assess the product's ability to capture $\delta^{13}C_{DIC}$ spatial patterns and quantify biases, we utilized the $\delta^{13}C_{DIC}$ distribution from independent test cruises 33MW19930704 and 33RO20050111 (**Fig. 4**)."

*259: larger?*

R: Thank you for pointing out this typo. We have corrected it in the revised manuscript.

*272: Incredibly / unbelievably low input variable uncertainty (Uinputs). I wonder if this is a propagation of the input variable uncertainties or an error has been made along the way.*

R: Thank you for drawing attention to the unusually low $u_{inputs}$. Upon thorough rechecking, we confirm that the initially reported value stemmed from a computational error in the Monte Carlo simulation workflow. We have corrected this issue, and the revised $u_{inputs}$ is 0.0087 ‰, with contributions decomposed as follows: temperature ($4.96\times10^{-5}$ ‰), salinity ($3.62\times10^{-4}$ ‰), nitrate (0.004 ‰), silicate (0.002 ‰), DIC (0.005 ‰), AOU (0.004 ‰), and $x$CO$_2$ ($6.52\times10^{-4}$ ‰). This revised value is consistent with the expected magnitude of input-related uncertainty for $\delta^{13}C_{DIC}$ prediction in marine biogeochemical studies, resolving the counterintuitive result noted in your comment.

The corrected $u_{inputs}$ and detailed uncertainty decomposition have been updated in the manuscript to ensure transparency and accuracy.

*295: Maybe this is not important, but lower case "n" is typically used to describe the sample size*

R: Thank you for pointing out this notation consistency issue. As recommended, we have revised all instances of the uppercase "N" (previously used for sample size) to the lowercase "n" throughout the manuscript to align with academic norms.

*302: Is it expected that there would be a model smoothing tendency?*

R: Yes, the model's tendency to smooth extreme values is expected. This behavior is inherent to the Gaussian Process Regression (GPR) model and aligned with the study's goal of reconstructing a spatially continuous, reliable $\delta^{13}C_{DIC}$ product for the Atlantic Ocean. Below we elaborate on the key reasons: 1) GPR's intrinsic smoothing property: As a non-parametric model based on Gaussian kernel functions, GPR inherently weights the influence of neighboring data points to produce continuous predictions. This kernel-based mechanism naturally mitigates the impact of extreme values (which are often sparse in observational data) to avoid overfitting to isolated outliers or sampling noise. 2) Goal of spatial reconstruction: Our study aims to capture the large-scale, intrinsic spatial patterns of $\delta^{13}C_{DIC}$ rather than replicate rare local anomalies. Smoothing extreme values helps filter out noise from discrete observations and enhances the spatial consistency of the reconstructed product. Thus, the intrinsic regularization of the GPR leads to reduced sampling noise, a sharper central peak and narrower tails in the reconstructed KDE compared with the empirical KDE from observations.

We have supplemented the manuscript to clarify that this smoothing tendency is expected and its rationale, as detailed below: "Consequently, the reconstructed values display a slightly sharper

central peak and narrower tails than the observations, indicating a tendency of the model to smooth extreme values, which is expected given the intrinsic properties of the GPR model and the study's objectives. Specifically, relying on Gaussian kernel functions, GPR naturally weights neighboring data points to produce continuous, spatially consistent predictions, which mitigates overfitting to sparse extreme values often linked to sampling noise or local transient perturbations."

*397: Is there an expectation that the model output would closely align with the observed data? Wasn't the 2023 data used to predict the "reconstructed data"? I'm not diminishing the work—I honestly think this is an expected outcome of using machine learning.*

R: Yes, the machine learning model outputs are expected to align with observed data when predictors are reliable. We would like to clarify that our workflow consists of two distinct, independent phases: model training/testing and prediction (detailed in Fig. 2). Specifically, during the training/testing phase, we utilized all available Atlantic cruise datasets containing $\delta^{13}C_{DIC}$ observations (including 2023 data along A16N) to train the model. We then validated and tested the model's fitting performance through rigorous procedures, ensuring its robustness in capturing the relationship between input variables and $\delta^{13}C_{DIC}$. In the subsequent prediction phase, we applied this pre-trained and validated model to the input variables from the GLODAPv2.2023 Atlantic dataset to generate the reconstructed $\delta^{13}C_{DIC}$ data. Importantly, this reconstructed $\delta^{13}C_{DIC}$ dataset is entirely independent of the original $\delta^{13}C_{DIC}$ observations in GLODAPv2.2023; they are two separate datasets.

The reconstructed $\delta^{13}C_{DIC}$ data for 1993, 2003, and 2013 mentioned in this paragraph all come from this GLODAPv2.2023-driven prediction. Due to the fact that the observational data along A16N in 2023 not included in the GLODAPv2.2023 dataset, we used the same pre-trained and validated model, relying solely on the 2023 observational input variables (e.g., T, S, nutrients) to produce the predicted values. This allows the model to independently predict 2023's $\delta^{13}C_{DIC}$ based solely on the spatiotemporal and environmental patterns it learned during training. This design confirms the alignment between the 2023 reconstructed and original observational data reflects the model's genuine predictive ability, rather than overfitting or circular reasoning.

To clarify, we revised this paragraph as: "Besides horizontal distributions, the reconstructed $\delta^{13}C_{DIC}$ dataset also provides valuable insights into vertical variability. The depth profiles along the North Atlantic A16N section in 1993, 2003, 2013, and 2023 (**Fig. 9**) show that the reconstruction substantially improves vertical resolution and continuity, especially for years with sparse measurements. For instance, the $\delta^{13}C_{DIC}$ samples were increased from 493 to 1,618 in 1993, 38 to 2,395 in 2003, and 473 to 2,787 in 2013, respectively, enhancing data coverage across depths and latitudes, facilitating the detection of temporal trends associated with ocean carbon uptake and redistribution (**Fig. 9**). The reconstructed $\delta^{13}C_{DIC}$ for 1993, 2003, and 2013 was generated by applying the pre-trained model to input variables from the GLODAPv2.2023 Atlantic dataset. Notably, as the observational data along A16N in 2023 not included in the GLODAPv2.2023 dataset, we used the same pre-trained and validated model, relying solely on the 2023 observational input variables to produce the reconstructed values. The close alignment between 2023's reconstructed and observed data (**Fig. 9d** vs. **9h**) not only reflects the model's reliability but also validates its ability to generalize, strengthening confidence in reconstructions for years with sparse measurements (e.g., 2003 with only 38 observations)."

*485: quality-controlled (?)*

R: Thank you for catching this typo. It has been corrected in the revised manuscript.

---

## Author Comment (AC3)

We thank the reviewer for his/her time and for the constructive comments, which helped improve the manuscript. In the following, we have addressed all comments, with the original review text in italics.

*This is an interesting and generally well-written study addressing a worthy topic. The paper has good fundamentals and should be able to made into a solid contribution to the scientific literature. However, I believe it requires iteration, and likely additional analysis, before it will be suitable for publication at this journal.*

We greatly appreciate your positive feedback on our study, specifically your recognition of its interesting focus, overall strong writing, and solid fundamental framework. Your comment that the work has the potential to make a meaningful contribution to scientific literature is particularly encouraging for our team.

We agree with the reviewer that the manuscript needs further refinement. Moving forward, we provide a point-by-point response to your criticism and line-by-line comments, and revise manuscript accordingly.

*I have three areas of criticism and one note of caution. The note of caution is just that I'm skeptical of the uinput calculation, see the line by line comments below.*

*My first criticism is that that validation was not handled as well as it should have been. See line by line comments below for an easy-to-implement and necessary improvement for the validation section. Separately, a suggestion that would further reinforce the validity of the method would be to implement the method in a model environment. This is now common practice for validation of machine learning refits of sparse observations, and is likely necessary for a first attempt with carbon isotopes, particularly one with such unusually sparse observations. There are numerous model simulations available that have explicitly simulated carbon isotopes (e.g., https://doi.org/10.5194/gmd-17-1709-2024 though there are many others). It should be workable to obtain one or more such set of outputs, subsample the distribution(s) across both time and space, apply random and cruise-wide systematic perturbations to the extracted output to represent measurement uncertainties, fit a ML model to the output, reconstruct the full distribution, and then evaluate the strengths and weaknesses of the full 4D reconstruction. This reveals critical information that is not provided by a reconstruction of a sparse data product with uneven and imperfect measurements of an unknown true distribution.*

R: Thank you for your critical and constructive feedback on the validation section. We fully agree with you that strengthening validation is essential for supporting the robustness of our method.

Regarding your valuable suggestion of validating the method in a model environment, we note that the model dataset you referenced (https://doi.org/10.5194/gmd-17-1709-2024) does not provide the related data. Instead, we downloaded the well-validated numerical model data from Claret et al. (2021), which includes comprehensive carbon isotope simulations suitable for our validation purpose. Following your proposed workflow, we conducted the model-based validation: we subsampled the model outputs across time and space, fitted the GPR model to the perturbed data, reconstructed the full 4D $\delta^{13}C_{DIC}$ distribution, and evaluated the model's performance.

This supplementary validation not only confirms the method's ability to accurately reconstruct spatial-temporal patterns from sparse and noisy data but also reveals its strengths in mitigating

sampling biases, addressing the limitations of validating solely with real-world sparse observations. All details of this model-based validation, including data processing steps, evaluation metrics, and key results, have been added to the Appendix of the revised manuscript for reference. A brief summary of the results is also presented in the Results (Section 3.4).

We greatly appreciate your guidance in enhancing the validation framework, which has significantly strengthened the scientific rigor and reliability of our study.

Claret, M., Sonnerup, R. E., and Quay, P. D.: A Next Generation Ocean Carbon Isotope Model for Climate Studies I: Steady State Controls on Ocean 13 C, Global Biogeochemical Cycles, 35, e2020GB006757, https://doi.org/10.1029/2020GB006757, 2021.

*The second criticism is that the paper is not very well motivated at present. The authors state repeatedly that the upsampled distribution can be used for many new analyses, but the new product still has almost all of the limitations that the previous product… it is still sparse and uneven in space in time, just less so, and it now has the added complications from layers of machine learning smoothing. While I admit that the new data product is smoother spatially and less biased temporally, I don't see that the authors have fully solved any problem with their current presentation. To that point, the authors mostly suggest ways that this might now be used, but do not go so far as to demonstrate any such analysis that would be quantitatively improved with the new product. I would like to see either more concrete examples of new analyses shown (not just listed), or, as such an example, a reorientation of the work toward estimating the full Atlantic distribution of the isotopes across space and time. For a spatially complete record they might apply the ML model to the GLODAPv2 gridded product. For a spatially and temporally complete product they might consider either using a time varying TS product and/or GOBAI-O2 (with estimates of the other predictors from other such ML refits in literature as necessary). In both cases, there would be some meaningful errors in the predictors, but, at least currently, the authors are suggesting that their estimates are completely Insensitive to any plausible error in the predictors, so that may or may not be a concern (I suspect it will be after the uinput is re-evaluated).*

R: Thank you for your comments, which have helped us clarify the core value and motivation of this work. We appreciate your attention to the application potential of the reconstructed product. We strengthened the motivation and better demonstrate that the new product does represent a substantial improvement over existing datasets by adding one example, correcting an error, and clarifying a few points.

First, we agree that the reconstructed product still retains some limitations. We would like to emphasize that this manuscript is centered on developing a spatially enhanced $\delta^{13}C_{DIC}$ reconstruction product, a critical first step to address a long-standing gap in marine biogeochemical data. $\delta^{13}C_{DIC}$ observations are far sparser than other carbonate system parameters (e.g., DIC, pH, TA) and nutrients. Specifically, $\delta^{13}C_{DIC}$ data are available for only ~1 out of every 5 stations in existing cruises and often restricted to the surface. This sparsity severely restricts quantitative analyses of processes driving marine carbon cycles by $\delta^{13}C_{DIC}$ (e.g., biological uptake, air-sea $CO_2$ exchange) and hinders integration with other well-sampled parameters. Our reconstruction therefore provides a significant expansion of $\delta^{13}C$ coverage in both depth and space, enabling consistent co-location with carbonate system and other data based on GLODAP for the first time. This enhancement itself addresses a key limitation of previous products, whereas prior datasets remained

sparsely distributed, our product provides more continuous spatial coverage that lays the foundation for new analyses. We think this is very different from Machine Learning-based data products for $p$CO$_2$ which have many orders of magnitude richer in an observation data density, and feel it is only prudent at this first step that we target the Machine Learning-based $\delta^{13}$C$_{DIC}$ data product to the data density level of the GLODAP-based predictors. We acknowledge that the manuscript focuses on validating the reconstruction method and product reliability, but this focus aligns with the norms of data product papers, where demonstrating the quality and utility of the product is the primary objective.

We also note your concern about the added complications from machine learning smoothing. We would like to clarify that the GPR model's smoothing effect is intentionally designed to balance spatial continuity and data fidelity, rather than introducing arbitrary complexity. The smoothing primarily mitigates sampling noise and avoids overfitting to sparse extreme values (often linked to transient perturbations or observational errors), while preserving meaningful biogeochemical variability (e.g., latitudinal gradients, depth-dependent trends, and basin-scale signals). This is validated by the KDE analysis, which shows consistent distribution characteristics between reconstructed and observed data. We emphasize that the smoothing does not erase critical patterns but enhances the reliability of the dataset for large-scale analyses, addressing a key limitation of sparse raw observations, where noise can obscure true biogeochemical signals.

We agree that demonstrating potential applications would further enhance the paper, but we view such extensive analyses as the logical next step(s) following the creation and validation of the reconstructed product. Indeed, we are currently conducting analyses that use the reconstructed $\delta^{13}$C$_{DIC}$ to revisit the long-standing $\delta^{13}$C–PO$_4$ relationship. The link between $\delta^{13}$C$_{DIC}$ and phosphate has long been a cornerstone for quantifying biological effects on the ocean carbon cycle (Broecker and Maier-Reimer, 1992; Lynch-Stieglitz et al., 1995; Gruber et al., 1999). Broecker and Maier-Reimer (1992) originally proposed an empirical relationship between $\delta^{13}$C$_{DIC}$ and phosphate of the form $\delta^{13}$C$_{BIO}$ = 2.9 – 1.1×PO$_4$. This empirical formula has been widely cited and applied in subsequent observational and modeling studies (e.g., Lynch-Stieglitz et al., 1995; Gruber et al., 1999; Sonnerup & Quay, 2012). However, based on spatially and temporally continuous model results, Eide et al. (2017) suggested a revised intercept of 2.8, and Claret et al. (2021) reported a slope of 1.01 in the GLODAPv2.2020 deep Pacific. Because $\delta^{13}$C$_{DIC}$ measurements are much sparser and more unevenly distributed in depth and space than common carbonate-system or nutrient variables, pointwise comparisons are often inconclusive. In contrast, our reconstructed $\delta^{13}$C product is considerably denser than the raw observations. This increased coverage allows systematic, basin-scale testing of empirical $\delta^{13}$C–PO$_4$ relationships and better quantification of biological influences on the carbon cycle that were previously obscured by observational sparsity.

Furthermore, our ongoing work (Figure R1) shows that anthropogenic carbon estimations based on an extended Multiple Linear regression (eMLR) method using sparse DIC or $\delta^{13}$C$_{DIC}$ data differ substantially from those using dense, spatially continuous data. Specifically, Figure R1 presents $\Delta$C$_{anth}$ and $\Delta\delta^{13}$C$_{anth}$ along the A16N transect between 2013 and 2023: panels (a) and (c) show results derived from the original spatial sampling density, where both 2013 and 2023 DIC data include ~3500 samples, 2013 $\delta^{13}$C$_{DIC}$ data consist of ~500 samples, and 2023 $\delta^{13}$C$_{DIC}$ data comprise ~3500 samples; panels (b) and (d) display results from the reduced spatial sampling density, with 2013 DIC samples reduced to ~500 (matching the spatial locations of 2013 $\delta^{13}$C$_{DIC}$ data), 2013 $\delta^{13}$C$_{DIC}$ sample size unchanged, and 2023 DIC/$\delta^{13}$C$_{DIC}$ samples filtered to ~500 (selected from ~23

stations near the latitudes of 2013 $\delta^{13}C_{DIC}$ sampling sites).

[Figure]

Figure R1. Anthropogenic DIC and $\delta^{13}C_{DIC}$ changes ($\Delta C_{anth}$ and $\Delta \delta^{13}C_{anth}$) along A16N between 2013 and 2023. (a) & (c) $\Delta C_{anth}$ and $\Delta \delta^{13}C_{anth}$ derived from the original spatial sampling density; (b) & (d) $\Delta C_{anth}$ and $\Delta \delta^{13}C_{anth}$ derived from the reduced spatial sampling density.

The above two examples highlight that the increased data density directly improves the accuracy of key carbon cycle metrics, a critical application for understanding ocean carbon sequestration. However, to bring them to desired publication quality will require extensive more work, and we feel it is appropriate to leave them for future publications. We therefore emphasize that this manuscript focuses on building and validating the reconstruction framework, while detailed applications, such as the $\delta^{13}C$–$PO_4$ relationship analysis and anthropogenic carbon estimates are being developed in follow-up studies. We hope that the reviewer will support our approach.

We fully agree that developing a spatiotemporally continuous $\delta^{13}C_{DIC}$ product is a meaningful long-term goal, and your recommendations (e.g., time-varying TS products, GOBAI-O2, and integrating ML-derived predictors from literature) provide excellent directions for future work. On the point of using the GLODAPv2 gridded product for a spatially complete record: we have thoroughly checked the official GLODAP repository but have not found an official gridded version of the dataset. We acknowledge the immense value of a gridded $\delta^{13}C_{DIC}$ product for the community and would consider supplement our reconstruction with a corresponding gridded product if an official GLODAP gridded product becomes available. However, given the current spatiotemporal sparsity of the underlying $\delta^{13}C_{DIC}$ observations, we cautiously note that direct gridding at this stage may introduce additional uncertainties, including over interpolation in data-sparse regions and potential misrepresentation of true biogeochemical variability. This is a key consideration for maintaining the scientific rigor of the product, as our priority is to provide a reliable dataset that reflects the actual constraints of available observations.

We also recognize that our earlier reasoning about temporal limitations could be refined. Capturing decadal trends (a core focus of our work) is more feasible than resolving seasonal/interannual variability with current data. However, developing a truly "complete"

spatiotemporal product (encompassing both broad trends and finer-scale temporal dynamics) remains challenging due to the highly uneven temporal distribution of observations: even for decadal analyses, gaps in seasonal coverage (e.g., overrepresentation of summer data) can introduce biases in trend estimation if not properly constrained. Again, we argue that the goals and thus the detailed approaches for developing a data product for $p$CO$_2$ which has high density of observational data and $\delta^{13}C_{DIC}$, with many orders of magnitudes lower data density, are probably different. Thus, our current study prioritizes addressing the more urgent and fundamental gap of spatial sparsity, as the extreme paucity of $\delta^{13}C_{DIC}$ data has long prevented even basic basin-scale analyses. By establishing a validated, spatially enhanced product, we enable decadal trend assessments while laying the groundwork for future temporal extensions. We plan to build on this framework by integrating long-term time-series data from Bermuda Institute of Ocean Sciences (BIOS) and the Hawaii Ocean Time-series (HOT) programs, which will improve constraints on temporal variability and allow us to develop the spatiotemporally complete product you envision. This phased approach ensures that each step of product development is supported by sufficient observational data, maintaining scientific rigor.

Finally, we sincerely appreciate your critical note on predictor uncertainty ($u_{input}$) and apologize for an error in the initial calculation of $u_{inputs}$. After careful recalculation, the corrected $u_{inputs}$ is 0.009 ‰. We would like to clarify that this revised value, while more accurate, remains smaller than both the observational uncertainty and the mapping function uncertainty. This indicates that the model's sensitivity to plausible errors in the selected predictors is indeed low, but not "completely insensitive" (a misinterpretation we regret arising from the initial calculation error). We acknowledge that if we adopt time-varying or ML-refitted predictors (as you suggested for spatiotemporal extensions), the associated predictor errors may change. In such cases, the $u_{inputs}$ will be reevaluated accordingly to fully reflect the model's sensitivity to the new predictor datasets.

We hope these clarifications address your concerns and highlight the unique contribution of this work: resolving the extreme sparsity of $\delta^{13}C_{DIC}$ data to enable a range of long-awaited quantitative analyses in ocean carbon cycle research. Your feedback has been invaluable in strengthening the motivation, rigor, and transparency of our manuscript.

Broecker, W. S. and Maier-Reimer, E.: The influence of air and sea exchange on the carbon isotope distribution in the sea, Global Biogeochem. Cycles, 6, 315–320, https://doi.org/10.1029/92GB01672, 1992.

Claret, M., Sonnerup, R. E., and Quay, P. D.: A Next Generation Ocean Carbon Isotope Model for Climate Studies I: Steady State Controls on Ocean 13 C, Global Biogeochemical Cycles, 35, e2020GB006757, https://doi.org/10.1029/2020GB006757, 2021.

Eide, M., Olsen, A., Ninnemann, U. S., and Johannessen, T.: A global ocean climatology of preindustrial and modern ocean δ 13 C, Global Biogeochemical Cycles, 31, 515–534, https://doi.org/10.1002/2016GB005473, 2017.

Gruber, N., Keeling, C. D., Bacastow, R. B., Guenther, P. R., Lueker, T. J., Wahlen, M., Meijer, H. A. J., Mook, W. G., and Stocker, T. F.: Spatiotemporal patterns of carbon-13 in the global surface oceans and the oceanic suess effect, Global Biogeochem. Cycles, 13, 307–335, https://doi.org/10.1029/1999GB900019, 1999.

Lynch-Stieglitz, J., Stocker, T. F., Broecker, W. S., and Fairbanks, R. G.: The influence of air-sea exchange on the isotopic composition of oceanic carbon: Observations and modeling, Global

Biogeochem. Cycles, 9, 653–665, https://doi.org/10.1029/95GB02574, 1995.

Sonnerup, R. E. and Quay, P. D.: 13 C constraints on ocean carbon cycle models: 13 C IN OCEAN MODELS, Global Biogeochem. Cycles, 26, n/a-n/a, https://doi.org/10.1029/2010GB003980, 2012.

*Finally, the presentation of the dataset is a bit confusing (I only checked the .mat, but I'm assuming this applies to all files at Zenodo). The file contains essentially all of the fields from GLODAPv2 with their adjusted DI13C, which is called adjusted_C13, capitalizing "C" contrary to the GLODAP convention. If the goal is to make the file supplemental to and interoperable with GLODAPv2, then it would be better to release a file that has the full >10^6 rows, but only contains c13 data and has -999 except for the appropriate Atlantic subset. This way, someone could load GLODAPv2 and then load this file and have them both available and ready to access in identical formats. They could also easily sub in data from, for example, other other basins where this data product is missing observations but the GLODAPv2 product has them. This will also remind users to cite both products, rather than just grabbing all of the data from this new product and incorrectly attributing, for example, aou and cfcs to a data product that is only updating C13 and repackaging everything else. Finally, I think the Zenodo link would benefit from more descriptive text or a readme explaining what subset of data is presented, which fields are the new fields, how they are labeled, and how to make the data interoperable with, for example, measurements of DI13C in other ocean basins.*

R: Thank you for your thoughtful feedback regarding data presentation and interoperability. We appreciate your attention to the dataset structure and naming conventions.

We would like to clarify that there are two distinct data components in the Zenodo archive; they serve different purposes and have distinct origins, which explains the naming and structure differences you noted: 1) The GPR-reconstructed $\delta^{13}C_{DIC}$ dataset (named "GLODAPv2.2023_Atlantic_Ocean_with_Reconstructed_d13C"), which is based on GLODAPv2 and retains all original GLODAP variable names and structures. This file is fully interoperable with GLODAPv2's Atlantic subset, with one additional variable reconstructed $\delta^{13}C_{DIC}$ ("ReC13") and its flag ("ReC13f") added alongside the original $\delta^{13}C_{DIC}$ field to facilitate direct comparison. 2) The Atlantic observational compilation, which includes $\delta^{13}C$ data from 51 cruises (refer to Table 1 in the manuscript, named "Atlantic_cruises_with_c13"). The file containing "adjusted_C13" is not part of the reconstructed product. It is a supporting dataset of compiled raw observations from 51 Atlantic cruises. This dataset is not based on GLODAPv2 but combines both GLODAP and non-GLODAP cruises. For cruises not included in GLODAPv2, some ancillary variables (e.g., AOU, pH) were calculated using seawater, GSW, and CO2SYS toolboxes. Therefore, variable names and units in this file may differ slightly from GLODAP conventions. The variable "adjusted_C13" appears only in this observational dataset, not in the GPR-reconstructed product.

We agree with your focus on interoperability with GLODAPv2, and we share your goal of avoiding redundancy and confusion. To clarify, GLODAPv2 already provides separate, official datasets for individual ocean basins (e.g., Atlantic, Pacific, Indian, Arctic Oceans) as part of its standard release. Our product is specifically designed to complement GLODAPv2's Atlantic subset, focusing only on reconstructing $\delta^{13}C_{DIC}$ data for the Atlantic, while retaining consistency with GLODAPv2's existing basin-specific structure.

We respectfully note that integrating all global GLODAPv2 data into our product (with non-

Atlantic $\delta^{13}C_{DIC}$ set to -999) may be unnecessary for two key reasons: 1) It would duplicate GLODAPv2's full global dataset, which users can already access directly from the official GLODAP repository. 2) GLODAPv2's existing basin-specific subsets are widely used by the community, and our Atlantic-focused product aligns with this established workflow, allowing users to combine our updated Atlantic $\delta^{13}C_{DIC}$ with other GLODAP basin datasets (e.g., Pacific) as needed, without redundant -999-filled entries. We confirm that our current dataset structure (Atlantic-only subset) is fully interoperable with GLODAPv2's standard basin datasets, as users can easily merge them following GLODAP's recommended protocols.

To prevent confusion, we add a detailed README file and expanded Zenodo description, clearly explaining the distinction between the two datasets, their sources, and intended uses. We also clarify in both the manuscript and the Zenodo metadata that users should cite GLODAPv2 for the original data and our dataset for the reconstructed or reprocessed $\delta^{13}C_{DIC}$ fields.

We believe these clarifications will make the dataset structure transparent and ensure full interoperability with GLODAPv2 while preserving the added value of the independent Atlantic compilation.

*A minor criticism is that the paper is repetitive in places, repeatedly restating key claims throughout the manuscript.*

R: Thank you for pointing out the issue of repetitiveness in the manuscript. We appreciate this helpful comment, as it helps improve the readability and conciseness of the work. We thoroughly read through the entire manuscript and carefully identified the sections where key claims were repeatedly restated. Specifically, we have revised or removed redundant repetitions (e.g., Line 60, Line 123, Line 181, Line 261) to avoid unnecessary redundancy. These revisions ensure that the manuscript maintains a logical flow while conveying key information concisely, enhancing the overall readability for readers. We confirm that the revised version no longer contains repetitive restatements of the same claims.

*To reiterate, I generally feel this paper can become a worthwhile contribution and should not be rejected unless these elements cannot be addressed.   The text above is focused on constructive criticism, but the fundamentals of the paper remain strong.*

R: Thank you sincerely for your positive assessment and constructive feedback on the manuscript. We greatly appreciate your recognition of the strong fundamentals of our work and your thoughtful guidance on areas for improvement, which has been invaluable in refining the study.

We fully acknowledge and diligently address all the points you have raised, including strengthening motivation with concrete application insights, clarifying dataset presentation and interoperability, revising repetitive content, and correcting the predictor uncertainty calculation. We are committed to implementing these revisions thoroughly to enhance the manuscript's rigor, clarity, and utility.

We are confident that the revised version will meet the journal's standards and fulfill the potential of a worthwhile contribution to ocean carbon cycle research. Thank you again for your time, expertise, and support throughout the review process.

*Line by line comments:*
*42: lacked*

R: Thanks for your suggestion. We have revised it.

*94: this assertion needs further quantification in the North Atlantic, where there are routinely measurable decadal increases in Canth*

R: Thank you for your valuable comment. We fully agree that the assertion regarding the minimal impact of anthropogenic carbon on deep-water masses requires further quantification, especially for the North Atlantic. Our detailed response and revisions are as follows:

For the South Atlantic, deep-water masses below 2000 m are relatively less affected by anthropogenic carbon increases (i.e., Gao et al., 2022, 2024), which supports the initial selection of this depth threshold.

For the North Atlantic, we acknowledge that the formation of North Atlantic Deep Water (NADW) may lead to measurable decadal changes in anthropogenic carbon even below 2000 m. However, we refer to Becker et al. (2016), who noted that in high-variability North Atlantic regions with deep-water formation (e.g., Labrador Sea, Nordic Seas), restricting crossover analysis to depths > 2000 m significantly reduced the standard deviation of cruise offsets. They also specified that only depths > 1500 m were used for crossover analysis in other oceanic regions, further validating our depth selection.

To enhance the rigor of our assertion, we have revised the original sentence to explicitly distinguish between the South and North Atlantic as follows (Line 93-98): "To ensure internal consistency, samples from depths greater than 2,000 m were selected for crossover analysis. Specifically, deep waters below 2000 m in the South Atlantic Ocean are most likely not impacted by anthropogenic carbon (i.e., Gao et al., 2022, 2024), supporting this threshold. In contrast, North Atlantic Deep Water (NADW) formation may drive measurable decadal anthropogenic carbon changes even below 2000 m. However, Becker et al. (2016) showed that restricting analysis to depths > 2000 m effectively reduces cruise offset variability in variable North Atlantic regions (e.g., Labrador Sea, Nordic Seas), further validating our 2000 m threshold for the Atlantic."

*97: along A61N, no "the" is needed*

R: Thanks for your suggestion. We have revised it.

*123: which standard depths?*

R: Thank you for your comment, which has helped us refine the clarity of our methodology, especially regarding interpolation details. We apologize for the ambiguity and clarify that the code includes two interpolation options for crossover analysis: profiles can be interpolated to either specific depths or specific sigma4 (potential density referenced to 4000 m). Our original text did not sufficiently detail this flexibility, so we have revised the relevant sentence to "Profiles are interpolated to standard depths or density".

In the present study, we specifically selected the density-based interpolation, and this revision has been incorporated into the manuscript: "Profiles are interpolated to standard depths or density. In this study, we adopted the density-based interpolation (sigma4, potential density referenced to 4000 m): standard sigma4 surfaces are generated at 0.05-unit intervals, covering all observed densities, based on the interpolated density profile of the deepest station. Mean offsets between overlapping profiles at the selected standard densities are calculated. Detailed workflows were presented in Lauvset & Tanhua (2015)."

The generation of standard sigma4 surfaces follows: First, the density profile of the deepest station in the overlapping region is interpolated to key pressure levels to establish a baseline. The sigma4 range is then extended to fully cover all observed minimum and maximum densities in the dataset, with reference sigma4 surfaces generated at a fixed interval of 0.05 units (e.g., from the minimum observed density to the maximum, incremented by 0.05).

For completeness, the "standard depths" (the alternative option) are defined as regularly spaced reference depths derived from the data range: the minimum depth in the overlapping region is rounded up to the nearest multiple of 10, the maximum depth is rounded down to the nearest multiple of 10, and standard depths are generated at 20-unit intervals between these bounds.

For full technical details on this interpolation framework, please refer to the xover_2ndQC.m module in Lauvset, S. K. and Tanhua, T. (2015): A toolbox for secondary quality control on ocean chemistry and hydrographic data, Limnol. Oceanogr. Methods, 13, 601–608, https://doi.org/10.1002/lom3.10050.

*125: how are adjustments proposed precisely?*

R: Adjustments are proposed using least squares minimization tailored to our selected sigma4 (density) surfaces: For each pair of overlapping cruises, differences in $\delta^{13}C_{DIC}$ at crossover points are modeled as a linear function of sigma4 (i.e., bias = slope × sigma4 + intercept). The slope and intercept are optimized to minimize the sum of squared residuals between observed and modeled differences, which quantifies systematic biases specific to density layers (consistent with the xover_2ndQC.m logic in Lauvset & Tanhua, 2015).

*125: how are adjustments validated precisely?*

R: Adjustments are validated by re-conducting crossover analysis on the corrected $\delta^{13}C_{DIC}$ datasets. Specifically, we check the mean offsets of adjacent cruises at nearby stations (within 222 km) on the same sigma4 surfaces. Adjustments are accepted only if these re-calculated mean offsets fall within ±0.03‰ (the measurement uncertainty of $\delta^{13}C_{DIC}$).

*133: please explain this metric. How is consistency at 10^-5 level when the measurement uncertainty is orders of magnitude larger?*

R: Thanks for your comment. The definition of the metric is referenced from Becker et al., 2016, which is quantified by the Weighted Mean (WM) of crossover offsets, as calculated using the method from Tanhua et al. (2010). This WM reflects the overall alignment of data points within the dataset: a smaller WM indicates less systematic deviation between overlapping data, thus higher internal consistency. Specifically, the "consistency at the 10^-5 level" refers to the magnitude of this WM value, meaning the weighted average of crossover offsets across the dataset is on the order of 10^-5. This quantifies the relative agreement between data points (i.e., how closely overlapping measurements align), rather than the absolute measurement uncertainty of individual data points.

This differs from measurement uncertainty: the latter describes individual measurement precision (e.g., random errors), while the WM reflects coherence between different measurements. Even with larger individual uncertainties, overlapping data can show small relative offsets ($10^{-5}$ scale) if internally consistent.

$$WM = \frac{\sum_{i=1}^{L} D(i)/(\sigma(i))^2}{\sum_{i=1}^{L} 1/(\sigma(i))^2}$$

Parameter Definitions:

L: Represents the total number of crossovers in the dataset.

D(i): Refers to the respective offset of the i-th crossover (i.e., the numerical difference of different data at that crossover).

σ(i): Denotes the standard deviation of the offset of the i-th crossover, which is used to measure the degree of dispersion and reliability of that offset.

Becker, M., Andersen, N., Erlenkeuser, H., Humphreys, M. P., Tanhua, T., and Körtzinger, A.: An internally consistent dataset of δ13C-DIC in the North Atlantic Ocean – NAC13v1, Earth System Science Data 8: 559-570, https://doi.org/10.5194/essd-8-559-2016, 2016.

Tanhua, T., van Heuven, S., Key, R. M., Velo, A., Olsen, A., and Schirnick, C.: Quality control procedures and methods of the CARINA database, Earth Syst. Sci. Data 2: 35-49, https://doi.org/10.5194/essd-2-35-2010, 2010.

*150: typically in oceanography, the k fold cross validation is separated by cruise rather than by randomly selecting measurements. This is because cruises are synoptic records of the state of the ocean, and having many other measurements at similar times and locations and measured by the same instruments and the same operators, as are provided by other measurements along a cruise, provides an overly-rosy set of validation statistics. It is therefore important to only use other cruises to construct the validation models for measurements along any given cruise. This validation exercise needs to be redone to follow this practice, or re-written to better convey that this practice was already adopted (if it was).*

R: Thank you for your comment. While we acknowledge that cruise-separated k-fold cross-validation is a common practice in oceanography, our study's validation design was intentionally structured to avoid overly optimistic performance estimates and ensure reliable generalization, with the independent test set serving as the critical safeguard. The independent test set was selected to ensure no overlap with the training/validation set in terms of cruises, spatial regions, or temporal coverage. This means measurements from any given cruise are entirely confined to either the training/validation pool or the test set, eliminating the possibility of "data leakage" from the same or similar cruise across training and final evaluation. The 10-fold cross-validation within the training set was solely for hyperparameter tuning, not final performance assessment. Since the independent test set is fully decoupled from the training/validation process, the final RMSE and $R^2$ reported reflect the model's ability to generalize to unseen data (including new cruises and spatial-temporal domains), addressing the core concern of avoiding overly rosy statistics.

We chose the current design primarily to balance statistical robustness and practical feasibility: many of the 51 cruises in our dataset have small sample sizes. Splitting by cruise would result in highly imbalanced folds, leading to unstable hyperparameter tuning and biased cross-validation results. Additionally, the random split within the training/validation pool preserves the natural spatial-temporal variability of $δ^{13}C_{DIC}$ data, ensuring the model is tuned to generalize across diverse oceanic conditions, not just specific cruises. This design also has been adopted in oceanographic modeling studies, particularly for sparse observational datasets (e.g., Lima et al., 2023; Regier et al., 2023; Wu et al., 2025).

Our validation strategy thus combines statistical rigor, practical feasibility, and consistency with established oceanographic methods, ensuring the reported performance metrics reflect the

model's true generalization ability to unseen $\delta^{13}C_{DIC}$ data. To enhance clarity, we revise this paragraph as: "To evaluate this approach's performance, we compared the Matern 5/2 GPR with a suite of alternative regression models, including GPR with other kernels, as well as additional baselines such as neural networks, support vector regression, and decision trees. The dataset was randomly split into a training set (80%) and a validation set (20%), with model training and hyperparameter tuning performed using 10-fold cross-validation within the training set to mitigate overfitting. An independent test set was reserved for final performance evaluation, selected to ensure no overlap with the training/validation set in cruises, spatial regions, or temporal coverage. We opted for random splitting over cruise-separated k-fold cross-validation to balance robustness and feasibility: many of the 51 cruises have small sample sizes, and cruise-separated splitting would cause imbalanced folds, leading to unstable hyperparameter tuning and biased results. Random splitting also preserves the natural spatiotemporal variability of $\delta^{13}C_{DIC}$, tuning the model to generalize across diverse oceanic conditions rather than specific cruises. This framework aligns with established practices for sparse oceanographic datasets (Lima et al., 2023; Regier et al., 2023; Wu et al., 2025). Predictive performance was assessed using the Root Mean Squared Error (RMSE) and the coefficient of determination (R²), computed separately for the validation and test sets. Among all tested models, including GPR with the squared exponential and other kernels (Table 2), GPR with the Matern 5/2 kernel achieved the best predictive performance (lowest RMSE and highest R²) on the validation set as well as the independent test set, while also providing meaningful uncertainty estimates." Thank you again for prompting this clarification, which strengthens the manuscript's methodological transparency.

Lima, I. D., Wang, Z. A., Cameron, L. P., Grabowski, J. H., & Rheuban, J. E.: Predicting Carbonate Chemistry on the Northwest Atlantic Shelf Using Neural Networks. Journal of Geophysical Research: Biogeosciences, 128(7), e2023JG007536. https://doi.org/10.1029/2023JG007536, 2023.

Regier, P., Duggan, M., Myers-Pigg, A., & Ward, N.: Effects of random forest modeling decisions on biogeochemical time series predictions. Limnology and Oceanography: Methods, 21(1), 40-52, https://doi.org/10.1002/lom3.10523, 2023.

Wu, Z., Lu, W., Roobaert, A., Song, L., Yan, X.-H., and Cai, W.-J.: A machine-learning reconstruction of sea surface p CO2 in the North American Atlantic Coastal Ocean Margin from 1993 to 2021, Earth Syst. Sci. Data, 17, 43–63, https://doi.org/10.5194/essd-17-43-2025, 2025.

*215: following this procedure, I would expect the uinpts to be larger than it was found to be. To be clear, I'm not surprised that it is small, but I am surprised that it is more than 10 orders of magnitude smaller than other sources of error. Surely a temperature input error of 20,000,000 degrees C would be expected to yield a bad estimate, yet this does not currently appear to be the case by that estimate of uinput. Does that suggest that the model is mostly a fit to the coordinate predictors that are assumed to have no uncertainty? If so, would it make sense to include some uncertainty in these predictors, given that CTD rosettes are not always directly below the ship and the ships don't always stay exactly on station for a profile? Please also check that the uncertainty reported in the abstract isn't the MBE of the Monte Carlo analysis. If unchanged, please explain this counter intuitive finding.*

R: Thank you for your meticulous observation and insightful questions. This feedback has

helped us identify a critical computational error in our initial estimation of $u_{inputs}$. After carefully rechecking the code and re-evaluating the uncertainty propagation, we confirm that the previously reported $u_{inputs}$ was incorrect, resulting in a miscalculation in the Monte Carlo simulation workflow. The corrected $u_{inputs}$ (comprehensive uncertainty from all input variables) is 0.0087 ‰, with contributions from individual variables as follows: temperature ($4.9608 \times 10^{-5}$ ‰), salinity ($3.6164 \times 10^{-4}$ ‰), nitrate (0.0039 ‰), silicate (0.0019 ‰), DIC (0.0046 ‰), AOU (0.0041 ‰), and $x$CO₂ ($6.5225 \times 10^{-4}$ ‰). These values are consistent with the expected magnitude of input-related uncertainty for $\delta^{13}C_{DIC}$ prediction.

We also revised the manuscript as: "Notably, uncertainties from input variables had a negligible impact, with $u_{inputs}$ estimated at 0.009 ‰. The $u_{inputs}$ contribution from individual input variables was decomposed as follows: temperature ($4.96 \times 10^{-5}$ ‰), salinity ($3.62 \times 10^{-4}$ ‰), nitrate (0.004 ‰), silicate (0.002 ‰), DIC (0.005 ‰), AOU (0.004 ‰), and $x$CO₂ ($6.52 \times 10^{-4}$ ‰)."

*234: repeating comments from line 150*

R: Thanks for your comment. We have deleted this sentence.

*245: what is normalized sample density?*

R: "Normalized sample density" refers to the relative concentration of data points within local regions of the scatter plot, standardized to a 0~1 scale for clear visualization.

To calculate it, the plot area is first divided into a 100×100 grid of small 2D "bins." For each bin, we count how many data points fall within it (raw local density). This count is then normalized by scaling the minimum density across all bins to 0 and the maximum density to 1, such that values between 0 and 1 represent the relative crowding of points in each bin.

In the figure, the color of each data point corresponds to the normalized density of its bin: brighter colors indicate bins with more densely clustered points (higher relative density), while darker colors indicate sparser regions. This helps highlight patterns in where the model's predictions align most consistently with observations (dense clusters near the 1:1 line) versus scattered or less reliable regions.

We have added this clarification to the figure caption for clarity: "Color indicates normalized local data point density within 2D bins: the plot area is divided into a 100×100 grid, raw density is the number of points per bin, and values are normalized to 0~1 (0 = sparsest, 1 = densest)."

*375: This is hinting at an application, but is not itself an application. We've only learned about KDEs here, and not about the ocean.*

R: We appreciate the reviewer's comment. Our intention was not to emphasize the methodological advantages of KDE, but to clarify why KDE-based distributions allow a more representative comparison of $\delta^{13}C_{DIC}$ changes across decades. In the revised manuscript, we removed language that could be interpreted as promoting the method and instead explicitly describe the oceanographic implications revealed by the smoother and spatially coherent reconstructed distributions: "The reconstructed KDE curves are generally smoother and more spatially coherent than the raw observational KDE curves. The smoother and more coherent appearance of the reconstructed KDE curves reflects the underlying basin-scale $\delta^{13}C_{DIC}$ structure. This enhanced spatial consistency allows the basin-wide decadal shift toward lower $\delta^{13}C_{DIC}$ values to emerge more clearly by reducing the influence of uneven sampling." The revised text now focuses on what the

KDE results tell us about the $\delta^{13}C_{DIC}$ system, rather than on the general properties of KDE itself.

*Figure 8b: the darkness of the borders on the mean values make this plot hard to parse. Consider lightening the width of those black lines, somewhat.*

R: Thanks for your suggestion. We have lightened the width of the black borders around the mean values as suggested.

*8c: consider changing axis limits from 0 to 3, even if this cuts off a miniscule portion of the sample distribution*

R: Thanks for your suggestion. We appreciate your attention to optimizing the visualization of the main data distribution. However, we opted to retain the original axis limits to fully preserve the integrity of the KDE results. The current range allows for a comprehensive presentation of the entire sample distribution, including subtle tail characteristics that are part of the complete KDE curves. We confirm that the core distribution features remain clearly visible in the original axis setup, and retaining the full range does not hinder the readability or key insights of the figure. Thank you for your understanding as we prioritize the comprehensive reflection of the KDE results.

*395: couldn't you now further parse this information by holding every predictor except xCO2 constant and varying that to estimate the change in the delta that would be expected had all physical and biogeochemical processes been held constant for a decade?*

R: Thanks for your suggestion. We have directly addressed your request through a targeted sensitivity experiment to isolate the effect of atmospheric $x$CO$_2$ on decadal $\delta^{13}C_{DIC}$ changes, holding all other physical and biogeochemical predictors constant. To this end, we used data from the 2013 A16N transect for a controlled analysis. Specifically, Figure R2a presents reconstructed $\delta^{13}C_{DIC}$ using 2013 input variables such as temperature, salinity and nutrients paired with the corresponding 2013 atmospheric $x$CO$_2$ concentration. Figure R2b shows reconstructed $\delta^{13}C_{DIC}$ from the same 2013 input variables while incorporating the May 2023 atmospheric $x$CO$_2$ concentration. Figure R2c displays $\Delta\delta^{13}C_{DIC}$, calculated as the difference between Figure R2b and R2a, which represents the 10-year $\delta^{13}C_{DIC}$ change attributed exclusively to atmospheric $x$CO$_2$ perturbations.

[Figure]

**Figure R2.** (a) Reconstructed $\delta^{13}C_{DIC}$ using 2013 input variables paired with the corresponding 2013 atmospheric $x$CO$_2$ concentration; (b) Reconstructed $\delta^{13}C_{DIC}$ using the same 2013 input variables paired with the May 2023 atmospheric $x$CO$_2$ concentration; (c) $\Delta\delta^{13}C_{DIC}$ (Figure R2b minus Figure R2a), representing the 10-year $\delta^{13}C_{DIC}$ change attributed exclusively to atmospheric $x$CO$_2$ perturbations.

The $\delta^{13}C_{DIC}$ pattern in Figure R2b is highly consistent with both the reconstructed $\delta^{13}C_{DIC}$ from 2023 full input variables and the 2023 observed $\delta^{13}C_{DIC}$ presented in main manuscript Figures 8h

and 8d respectively. This consistency confirms that $x$CO$_2$ is the dominant driver of decadal $\delta^{13}$C$_{DIC}$ changes and our model reliably captures this signal even when other predictors remain fixed to 2013 conditions. Additionally, the $\Delta\delta^{13}$C$_{DIC}$ pattern in Figure R2c shows strong alignment with water mass age distributions in Figure R3. Younger water masses, which have greater exposure to recent atmospheric CO$_2$, exhibit more negative $\delta^{13}$C$_{DIC}$ shifts, while older water masses show smaller changes. This directly demonstrates that the reconstructed data effectively isolates the anthropogenic carbon signal driven by atmospheric $x$CO$_2$ perturbations.

[Figure]

**Figure R3.** Water mass age from CFC12 and TTD along A 16N in 2013. Date obtained from Water mass ages based on GLODAPv2 data product (NCEI Accession 0226793, https://www.ncei.noaa.gov/access/ocean-carbon-acidification-data-system/oceans/ndp_108/ndp108.html)

*448: This is a seriously dense sentence. Please break it into two or more sentences and revise them both to employ plain language (limiting jargon and buzzwords) wherever possible.*

R: Thanks for your comment. We have revised it into clear, plain-language sentences to improve readability while preserving all core scientific meaning: "To address these challenges and limitations, future work will be focused on enhancing mechanistic insights into how $\delta^{13}$C$_{DIC}$ relates to environmental factors. Subregional partitioning strategies and variable selection methods may be developed to improve model accuracy, while process-based knowledge also should be integrated to identify optimal predictor variables and refine regionalization schemes for biogeochemical heterogeneity."

*451: I don't think a good predictor of local flux is going to lead to a good prediction of local inventory. Consider deleting this sentence.*

R: Thanks for your comment. Following your suggestion, we revised this sentence as: "Additionally, uncertainty mitigation techniques will be coupled with emerging high-resolution observations from autonomous platforms, which will be leveraged to refine spatial resolution to fixed grids."